# TokenCLIP: Token-wise Prompt Learning for Zero-shot Anomaly Detection

## Abstract

Adapting CLIP for anomaly detection on unseen objects has shown strong potential in a zero-shot manner. However, existing methods typically rely on a single textual space to align with visual semantics across diverse objects and domains. The indiscriminate alignment hinders the model from accurately capturing varied anomaly semantics. We propose TokenCLIP, a token-wise adaptation framework that enables dynamic alignment for fine-grained anomaly learning. Rather than mapping all visual tokens to a single, token-agnostic textual space, TokenCLIP aligns each token with a customized textual subspace that represents its visual characteristics. Explicitly assigning a unique learnable textual space to each token is computationally intractable and prone to insufficient optimization. We instead expand the token-agnostic textual space into a set of orthogonal subspaces, and then dynamically assign each token a subspace combination guided by semantic affinity, which jointly supports customized and efficient token-wise adaptation. To this end, we formulate dynamic alignment as an optimal transport problem, where all visual tokens in an image are transported to textual subspaces under the cross-modal similarity cost matrix. The marginal constraint and minimal cost objective of OT ensure sufficient optimization across subspaces and encourage them to focus on different semantics. Solving the problem yields a transport plan that adaptively assigns each token to semantically relevant subspaces. A topK masking is applied to further sparsify the plan and specialize subspaces for distinct visual regions. Extensive experiments demonstrate the superiority of TokenCLIP.

## 1 Introduction

Foundation Models (FMs) (Radford et al., 2021; Kirillov et al., 2023; Qwen et al., 2025) have shown the potential to generalize to unseen class semantics and domains. This breakthrough has driven the rapid development of downstream tasks that explore zero-shot capabilities by adapting FMs (Pang et al., 2021; Zhou et al., 2022; Khattak et al., 2023; Jeong et al., 2023; Zhou et al., 2024a; Gu et al., 2024b). Anomaly detection has also followed this trend, evolving from specialized models toward more generalized detection frameworks (Jeong et al., 2023; Zhou et al., 2024a; Chen et al., 2023b; Cao et al., 2024; Qu et al., 2025; Jiang et al., 2025; Zhou et al., 2024b; Gao et al., 2025).

A prominent line of research in this area involves adapting CLIP for zero-shot anomaly detection. These methods typically project either learnable (Zhou et al., 2024a; Gu et al., 2024b) or handcrafted text prompts (Jeong et al., 2023; Chen et al., 2023b) into a shared embedding space and align them with diverse visual features, capturing both global and local abnormalities. Despite their appeal, these approaches rely on a single textual space to indiscriminately align with all visual tokens, whether detecting a crack on a carpet or a tumor in a brain scan. As shown in Figure 1(a), this coarse alignment makes it difficult for the detection model to capture generalized anomaly semantics, as the token-agnostic textual space is forced to make a tradeoff between diverse semantic tokens. As a result, the model tends to favor common anomalies while compromising the rare anomaly semantics.

A natural approach is to assign each visual patch token to its own textual embedding space. However, this design introduces two major challenges. Challenge 1: High computational cost. For instance, a 518×518 image typically yields 1,369 patch tokens. Assigning a unique textual embedding to each token requires encoding the same number of distinct text prompts through the text encoder. This leads to significant computational overhead. Challenge 2: Underfitting of textual space due to

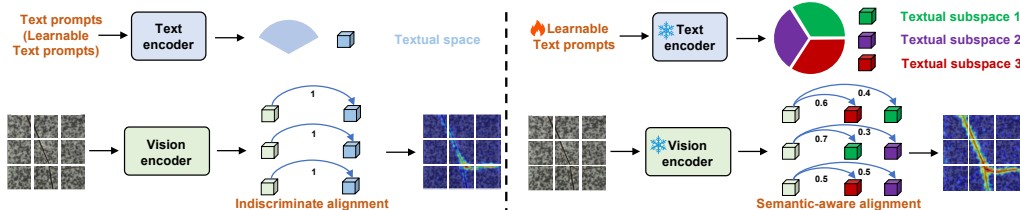

(a) Indiscriminate alignment used in previous works.    (b) Dynamic alignment in the proposed TokenCLIP.

Figure 1: Framework comparison. (a): Previous works rely on a textual space to indiscriminately align the diverse visual patch tokens, which would comprise the accurate alignment for diverse anomaly semantics. textbf(b): TokenCLIP introduces multiple orthogonal textual subspaces to dynamically align each visual patch token according to its visual semantics. This enables token-level textual supervision, resulting in fine-grained and comprehensive anomaly learning.

insufficient optimization. Since each token-specific textual embedding is updated only once during training, it results in severe underfitting and poor textual adaptation. To address these challenges, this paper proposes TokenCLIP, a fine-grained adaptation framework that dynamically aligns each visual patch token with a combinatorial set of orthogonal textual subspaces. Figure 1(b) shows that TokenCLIP enables token-level alignment based on visual semantics, thereby achieving granular modeling of anomaly semantics. TokenCLIP could circumvent the above challenges by 1) Leveraging weighted combinations of textual subspaces to provide token-level supervision avoids explicitly encoding individual textual embeddings for each token, thereby mitigating the computational burden; 2) Sharing orthogonal textual subspaces across all visual tokens enables sufficient optimization of each subspace and promotes their semantic specialization.

We formulate dynamic alignment as an optimal transport (OT) task, where all visual tokens in an image are transported to textual subspaces based on the cosine similarity between visual and textual representations. In this formulation, the visual patch tokens (source distribution) are required to be transported to the textual subspaces (target distribution), with minimal transport cost. The marginal constraint and minimal cost objective facilitate sufficient optimization and encourage semantic specialization across textual subspaces, respectively. Specifically, we first define a learnable text prompt to construct a base textual space, which captures global anomaly semantics through indiscriminate alignment. Building upon this space, we introduce a multi-head projection to derive multiple orthogonal textual subspaces, further regularized by an orthogonality constraint to promote semantic diversity. Unlike prior many-to-one alignment approaches, we model token-level alignment as a many-to-many correspondence between visual patch tokens and textual subspaces via OT. With visual-textual cosine similarity as the cost matrix, solving this OT problem yields a transport plan, where each entry quantifies the mass (weight) assigned from a visual token to a textual subspace. This formulation provides two key advantages: (1) marginal constraint of OT ensures that each textual subspace is sufficiently optimized; (2) minimization of total transport cost encourages textual subspaces to specialize in distinct visual semantics. We further sparsify the transport plan by retaining only the topK textual subspaces for each visual token. The selected masses are then normalized to produce soft assignment weights for final alignment. In doing so, TokenCLIP adaptively selects the most semantically relevant combination of textual subspaces for each token, without requiring an explicitly tailored textual space for every token. The main contributions of this paper are as follows:

- We reveal that current methods rely on indiscriminate alignment, which limits the capacity of textual spaces to capture comprehensive anomaly semantics. To address this limitation, we propose TokenCLIP, a novel fine-grained alignment framework that adaptively assigns a weighted combination of textual subspaces to each token. This enables semantics-aware textual supervision at the token level for fine-grained anomaly recognition.

- We formulate the dynamic alignment between tokens and orthogonal textual subspaces as an OT problem, where the marginal constraint and minimal cost objective respectively ensure sufficient optimization and encourage semantic specialization across subspaces. The transport plan is further sparsified by choosing topK subspaces for each token.

- We conduct extensive experiments across a wide range of object semantics to evaluate the effectiveness of TokenCLIP. Results on both industrial and medical benchmarks demonstrate its superiority in capturing diverse and comprehensive anomaly semantics.

## 2 RELATED WORK

**Zero-shot anomaly detection**    ZSAD is an emerging field that aims to detect anomalies in unseen object categories and even across domains (Esmaeilpour et al., 2022; Li et al., 2023; Gao, 2024). Some methods, such as VAND (Chen et al., 2023b) and AdaCLIP (Cao et al., 2024), focus on adapting CLIP's visual encoder to capture anomaly semantics. However, these approaches rely heavily on human-crafted text prompts to represent normality and abnormality (Jeong et al., 2023). Another branch, including methods like AnomalyCLIP (Zhou et al., 2024a), FiLo (Gu et al., 2024a) takes the opposite approach: rather than adapting the visual space, they learn text prompts to adapt the textual space for modeling anomaly semantics. Additionally, methods such as AACLIP (Ma et al., 2025), BayesCLIP (Qu et al., 2025), and AdaptCLIP (Gao et al., 2025) aim to adapt both the visual and textual spaces for improved performance. Despite these advances, most existing methods rely on a single textual space to simultaneously align with diverse visual patch tokens. FAPrompt (ZHU et al., 2025) ensembles multiple prompts to capture more abnormal patterns, but remains at the image level. In contrast, TokenCLIP introduces a dynamic alignment that provides token-level supervision by assigning each visual patch token to a semantics-aware weighted combination of textual subspaces.

**Prompt Learning**    Prompt learning was proposed to efficiently adapt CLIP for more accurate image classification with minimal computational overhead (Zhou et al., 2022). AnomalyCLIP (Zhou et al., 2024a) extends prompt learning to zero-shot anomaly detection, aiming to capture both local and global semantics for anomaly classification and localization Gu et al. (2024a). They typically model a token-agnostic textual space and overlook the semantic differences among local regions. In contrast, TokenCLIP introduces an orthogonal textual subspace and adaptively combines these subspaces to align each token according to its visual semantics. We formulate this dynamic alignment between visual regions and textual prompts as an OT problem. While PLOT (Chen et al., 2023a) also incorporates OT for prompt learning, it is primarily designed for image-level classification. In contrast, we leverage OT for pixel-level anomaly segmentation and image-level anomaly detection, achieving fine-grained and spatially aware alignment.

**Optimal Transport**    OT has emerged as a powerful framework for comparing probability distributions, with wide applications in computer vision (Villani et al., 2008). The entropic regularized OT proposed by Cuturi (Cuturi, 2013) significantly improved computational efficiency via the Sinkhorn algorithm, making OT feasible for large-scale learning. Subsequent works (Genevay et al., 2016; Peyré & Cuturi, 2019) have extended OT to stochastic, unbalanced, and mini-batch scenarios. In multi-modal learning, OT has been employed for fine-grained alignment between modalities (e.g., image patches and text tokens), enabling interpretable and structure-aware correspondence. In contrast to prior work Chen et al. (2023a), we are the first to introduce OT for fine-grained anomaly semantics learning, particularly in local visual anomaly detection. We observe that standard OT often results in overly dense transport plans and propose a topK selection mechanism to enforce cleaner, more discriminative alignments between visual and textual spaces.

## 3 METHODOLOGY

This paper introduces TokenCLIP, a fine-grained adaptation framework for accurate anomaly detection in Figure 2. The key insight of TokenCLIP is to move beyond indiscriminate visual-textual alignment by introducing a dynamic alignment mechanism, which provides token-level textual supervision for each visual patch token. The proposed TokenCLIP framework comprises two key modules: (1) a multi-head text prompt that projects the base textual space into multiple orthogonal subspaces; and (2) a dynamic alignment mechanism by solving the OT plan to assign each visual patch token to the semantically relevant textual subspace or their weighted combinations.

**Premiliary**    Given an auxiliary dataset $\mathcal{D} = \{x_1, x_2, \dots\}$, each image $x \in \mathbb{R}^{3 \times H_{\text{image}} \times W_{\text{image}}}$ is accompanied by an image-level label $y \in \mathbb{R}$ and a pixel-level annotation $S \in \mathbb{R}^{H_{\text{image}} \times W_{\text{image}}}$. The image encoder of CLIP encodes $x$ into a global image embedding $f \in \mathbb{R}^d$ and a set of visual patch tokens $V = \{v_i\}_{i=1}^N$. Given the text prompts corresponding to the normality class $n$ and abnormality class $a$, the text encoder produces the associated textual embeddings $g_n$ and $g_a \in \mathbb{R}^d$. The image-level anomaly score $P_a(g_a, f) \in \mathbb{R}$ and segmentation $S_a \in \mathbb{R}^N$ at index $i$ are given by:

$$P_a(g_a, f) = \frac{\exp\left(\langle g_a, f \rangle / \tau\right)}{\sum_{c \in \{n,a\}} \exp\left(\langle g_c, f \rangle / \tau\right)}, \quad S_a^{(i)} = S_a(g_a, v_i) = \frac{\exp\left(\langle g_a, v_i \rangle / \tau\right)}{\sum_{c \in \{n,a\}} \exp\left(\langle g_c, v_i \rangle / \tau\right)},$$

where $\tau$ is a temperature scaling factor, and $\langle \cdot, \cdot \rangle$ denotes cosine similarity.

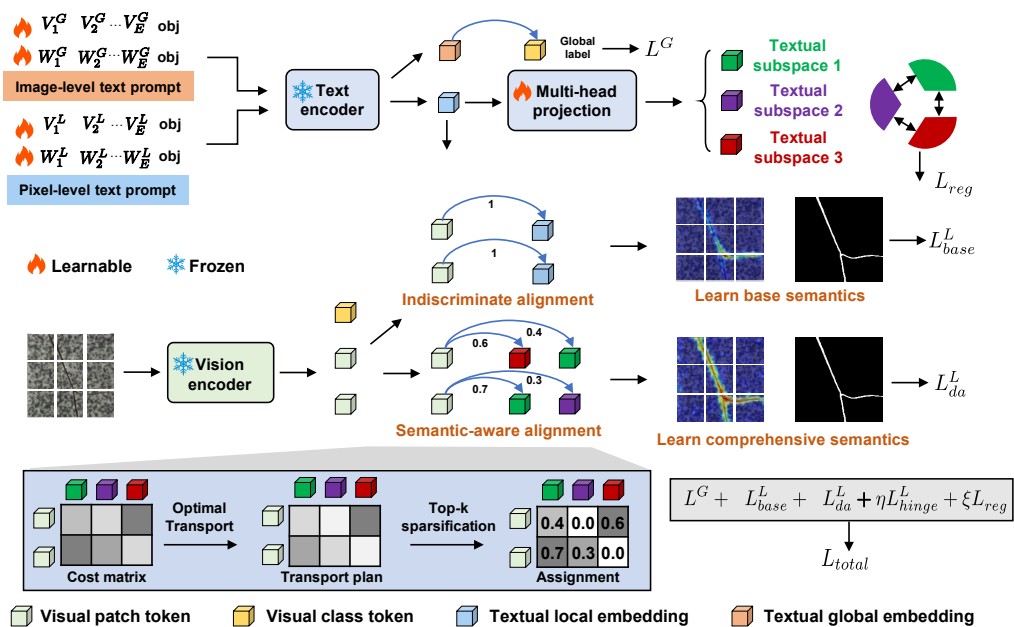

Figure 2: The Framework of TokenCLIP. TokenCLIP uses separate text prompts to learn global and local anomaly semantics. The global text embedding is aligned with the visual class token to detect image-level anomalies. In parallel, a corresponding local text prompt provides indiscriminate alignment with all visual patch tokens to capture base-level anomaly semantics. Building upon this, TokenCLIP employs a multi-head projection to map the base textual space into multiple orthogonal textual subspaces, regularized to encourage semantic diversity. The alignment between these subspaces and the visual patch tokens is then formulated as an OT problem. The resulting transport plan is sparsified as a token-wise assignment of textual subspaces. Finally, we jointly optimize TokenCLIP through end-to-end learning.

## 3.1 MULTI-HEAD TEXT PROMPT LEARNING

As mentioned above, the indiscriminate alignment suffers from performance sacrifice from using one textual space to match all visual semantics, including local and global abnormality. Therefore, we first use separate learnable text prompts, i.e., pixel-level text prompt and image-level text prompt, to model the local and global anomaly semantics. Formally, we define the pixel-/image-level prompt as $G = \{G_n, G_a\}$ and $L = \{L_n, L_a\}$. Each of them contains normality and abnormality prompts. Following AnomalyCLIP, we use the object-agnostic text prompt as follows:

$$G_n = [V_1^G] \cdots [V_E^G][object], \quad G_a = [W_1^G] \cdots [W_E^G][damaged][object].$$

$$L_n = [V_1^L] \cdots [V_E^L][object], \quad L_a = [W_1^L] \cdots [W_E^L][damaged][object],$$

where $V$ and $W$ are learnable word embeddings, and $E$ is the length of the learnable embedding. Using separate text prompts can help decouple local and global anomaly semantics. However, this separation would not be absolute, as local anomaly semantics can enhance global anomaly recognition. Motivated by this, we incorporate local anomaly semantics into the global text prompt by concatenating $G_n \in \mathbb{R}^{E \times D}$ and $L_n \in \mathbb{R}^{E \times D}$ along the channel dimension to form the final image-level text prompts $\bar{G}_n \in \mathbb{R}^{E \times D}$: $\bar{G}_n = \text{MLP}([G_n, L_n])$. MLP denotes a multi-layer perceptron, and $[\cdot, \cdot]$ represents the concatenation. The same operation is applied to $G_a$: $\bar{G}_a = \text{MLP}([G_a, L_a])$.

The text encoder encodes $\bar{G}_n$ and $\bar{G}_a$ to derive both global embeddings $\bar{g}_n$ and $\bar{g}_a$, and encodes $L_n$ and $L_a$ to derive local embeddings $l_n$ and $l_a$. We capture global anomaly semantics by matching the global textual embedding with the visual class token. The global loss $L_g$ is computed via cross-entropy as follows:

$$L_g = \text{CrossEntropy}([P_n(\bar{g}_n, f), P_a(\bar{g}_a, f)], y). \tag{1}$$

Inspired by AnomalyCLIP, $l_n$ and $l_a$ are indiscriminately aligned with all visual patch tokens $\{v_i\}_{i=1}^N$. Although this indiscriminate alignment fails to model each patch token precisely, this coarse-grained

modeling can provide a base anomaly semantics. To mitigate imbalance and promote the anomaly boundary, we combine Focal loss and Dice loss to learn the base anomaly semantics. Formally, the base local loss $L_{base}$ is given by:

$$L_{base} = \text{Focal}(\text{Up}([S_n, S_a]), S) + \text{Dice}(\text{Up}(S_a), S) + \text{Dice}(\text{Up}(S_n), I - S), \quad (2)$$

where $S_n^{(i)} = S_n(l_n, v_i)$ and $S_a^{(i)} = S_a(l_a, v_i)$ represent the $i$-th element of $S_n$ and $S_a$, respectively, and $\text{Up}(\cdot)$ denotes bilinear interpolation for upsampling.

Building on the base textual space, we further construct more fine-grained textual spaces. Specifically, we apply a multi-head projection to project the base embeddings into multiple fine-grained embeddings, i.e., $O_n = \{o_n^1, \cdots, o_n^j, \cdots, o_n^Q\}$ and $O_a = \{o_a^1, \cdots, o_a^j, \cdots, o_a^Q\}$, where $o_n^j \in \mathbb{R}^d$ and $o_a^j \in \mathbb{R}^d$. The process can be formally defined as:

$$\{o_n^j\}_{j=1}^Q = \text{MultiHead}_n(l_n), \quad \{o_a^j\}_{j=1}^Q = \text{MultiHead}_a(l_a). \quad (3)$$

Each head is implemented as a single-layer multilayer perceptron (MLP). To encourage semantic diversity and minimize redundancy among subspaces, we impose an orthogonality regularization:

$$\mathcal{L}_{\text{reg}} = \left\| [\tilde{o}_n^1, \cdots, \tilde{o}_n^Q]^\top [\tilde{o}_n^1, \cdots, \tilde{o}_n^Q] - I^{Q \times Q} \right\|^2 + \left\| [\tilde{o}_a^1, \cdots, \tilde{o}_a^Q]^\top [\tilde{o}_a^1, \cdots, \tilde{o}_a^Q] - I^{Q \times Q} \right\|^2, \quad (4)$$

where $\tilde{o}_n^j = o_n^j / \|o_n^j\|$ and $\tilde{o}_a^j = o_a^j / \|o_a^j\|$ are the $\ell^2$-normalized embeddings at $j$-th textual subspaces. We will elaborate on the fine-grained alignment between textual subspaces and visual patch tokens in the next section.

### 3.2 DYNAMIC ALIGNMENT VIA OT

Rather than applying a shared textual supervision to all visual tokens as in indiscriminate alignment, dynamic alignment aims to assign token-level supervision to better capture fine-grained anomaly semantics. However, explicitly providing a unique textual embedding for each patch token is computationally prohibitive due to the high cost of text encoding. To circumvent this, we propose to implicitly provide token-level supervision through a semantic-aware combination of textual subspaces. In this paper, we formulate the dynamic alignment as an OT problem between all visual patch tokens and the textual subspaces. The minimal cost objective enables each subspace to specialize in distinct anomaly semantics, while the marginal constraints ensure that all subspaces are sufficiently optimized. We consider visual embedding set $\{v_i\}_{i=1}^N$ and textual embedding set $\{o_c^j\}_{j=1}^Q, c \in \{n, a\}$ in the discrete formulation of OT. We define two empirical probability distributions supported on $\mathcal{V}$ and $\mathcal{O}$:

$$\boldsymbol{u} = \sum_{i=1}^N p_i \delta_{v_i}, \quad \boldsymbol{v} = \sum_{j=1}^Q q_j \delta_{o_c^j}, \quad (5)$$

where $\delta_{v_i}$ and $\delta_{o_c^j}$ denote Dirac delta measures centered at $\boldsymbol{v_i} \in \mathcal{V}$ and $\boldsymbol{o_c^j} \in \mathcal{O}$, respectively. The weights $p_i$ and $q_j$ are non-negative and satisfy $\sum_i p_i = \sum_j q_j = 1$ for marginal normalization.

The goal is to find a transport plan $T_c \in \mathbb{R}^{N \times Q}$ that minimizes the total transport cost between the two distributions. To reflect the affinity between visual and textual space, we compute the cosine distance between all visual and textual tokens to construct a cost matrix $\boldsymbol{C} \in \mathbb{R}^{N \times Q}$, where $\boldsymbol{C}_{ij} = 1 - \frac{v_i^\top o_c^j}{\|v_i\| \cdot \|o_c^j\|}$. However, solving this original problem is time-consuming due to the large computational complexity. We employ the Sinkhorn-Knopp algorithm to accelerate the solution of OT problems through entropic regularization.

$$\boldsymbol{T}_c^* = \min_{\boldsymbol{T}_c \in \Pi(\boldsymbol{u}, \boldsymbol{v})} \sum_{i=1}^N \sum_{j=1}^Q (\boldsymbol{T}_c \odot \boldsymbol{C})_{ij} - \lambda E(\boldsymbol{T}_c), \text{ subject to } \boldsymbol{T}_c \boldsymbol{1}^Q = \boldsymbol{u}, \ \boldsymbol{T}_c \boldsymbol{1}^N = \boldsymbol{v}, \quad (6)$$

where $\Pi(\boldsymbol{u}, \boldsymbol{v}) \in \mathbb{R}^{N \times Q}$ is the set of all joint distributions with marginals $\boldsymbol{u}$ and $\boldsymbol{v}$. The second term $E(\boldsymbol{T}_c) = \sum_{ij} (\boldsymbol{T}_c \odot \log \boldsymbol{T}_c)_{ij}$ is the entropy regularization, and $\lambda > 0$ is the regularization coefficient. This term encourages smoother and more numerically stable solutions. Moreover, the resulting optimization problem becomes strictly convex, enabling the OT plan to be computed in fewer iterations:

$$T_c^* = \text{diag}(\boldsymbol{u}^t) \exp\left(-\frac{\boldsymbol{C}}{\lambda}\right) \text{diag}(\boldsymbol{v}^t), \quad (7)$$

where $t$ is the iteration step, $\boldsymbol{u}^t$ and $\boldsymbol{v}^t$ are the scaling vectors updated via: $\boldsymbol{u}^t = \mu / \left(\exp(-C/\lambda)\boldsymbol{v}^{t-1}\right)$, $\boldsymbol{v}^t = \nu / \left(\exp(-C/\lambda)^\top \boldsymbol{u}^t\right)$, with the initialization $\boldsymbol{v}^0 = \boldsymbol{1}$. $\boldsymbol{T}^*$ is the plan minimizing the total cost, and it provides an underlying mapping for visual tokens to the given textual subspaces. Given the transport plan $T_c^*$, we retain only the topK entries in each row (i.e.,

per token), applying a threshold $\epsilon$ to filter out small values. This is because some low mass would interrupt the specialization of textual subspace learning. The selected values are then row-wise normalized as the affinities to serve as the final assignment matrix $A_c \in \mathbb{R}^{N \times Q}$, where each row is a sparse soft selection of textual subspaces for a visual patch token. Since affinity reflects the matching extent of visual space and textual space, the assignment is semantic-aware.

$$
A_c^{ij} = \begin{cases} (T_c^*)_{ij}, & \text{if } j \in \text{topK}\big((T_c^*)_{i,:}, k\big) \text{ and } (T_c^*)_{ij} > \epsilon, \\ 0, & \text{otherwise}, \end{cases} \quad \bar{A}_c^{ij} = \begin{cases} \dfrac{A_c^{ij}}{\sum_l A_c^{il}}, & \text{if } A_c^{ij} \neq 0 \\ 0, & \text{otherwise}. \end{cases} \quad (8)
$$

Holding the sparse assignment matrix $\bar{A} \in \mathbb{R}^{N \times Q}$, we can compute the logits for the class $c$. The final anomaly score for each visual patch token $v_i$ is calculated as:

$$
S_a^{\text{da}}(i) = \frac{\exp\left(z_a^i / \tau\right)}{\sum_{c \in \{n,a\}} \exp\left(z_c^i / \tau\right)}, \quad z_c^i = \begin{bmatrix} \bar{A}_c^{i1} & \bar{A}_c^{i2} & \cdots & \bar{A}_c^{iQ} \end{bmatrix} \cdot \big[\langle o_c^1, v_i \rangle, \langle o_c^2, v_i \rangle, \cdots, \langle o_c^Q, v_i \rangle\big]^{\text{T}}.
$$

This process allows each visual patch token to be dynamically aligned with a weighted combination of textual subspaces, enabling fine-grained and semantically aware cross-modal alignment. We introduce a dynamic alignment loss $L_{da}$ to achieve fine-grained anomaly modeling.

$$
L_{da} = \text{Focal}(\text{Up}([S_n^{da}, S_a^{da}]), S) + \text{Dice}(\text{Up}(S_a^{da}), S) + \text{Dice}(\text{Up}(S_n^{da}), I - S). \quad (9)
$$

**Theorem 3.1** (*OT penalizes subspace mixture and induces specialization*). *For $\ell_2$-normalized visual features $v_i$ and textual subspace $o_j$, we have $1 - \langle v_i, o_j \rangle = \frac{1}{2}\|v_i - o_j\|^2$, So the balanced OT objective becomes $\mathcal{L}_{\text{OT}} = \sum_{i=1}^N \sum_{j=1}^Q T_{ij} \|v_i - o_j\|^2, T \in \Pi(u, v)$. Let $\mathcal{C}_p$ and $\mathcal{C}_q$ be two visual clusters with centroids $\mu_p$ and $\mu_q$, and variances $\sigma_p^2 = \frac{1}{|\mathcal{C}_p|} \sum_{i \in \mathcal{C}_p} \|v_i - \mu_p\|^2$ and $\sigma_q^2 = \frac{1}{|\mathcal{C}_q|} \sum_{i \in \mathcal{C}_q} \|v_i - \mu_q\|^2$. If a subspace $o_j$ receives positive mass $\alpha_p > 0$ and $\beta_q > 0$ from $\mathcal{C}_p$ and $\mathcal{C}_q$, respectively, then its OT cost obeys*

$$
\underbrace{\sum_{i \in \mathcal{C}_p} T_{ij}\|v_i - o_j\|^2 + \sum_{i \in \mathcal{C}_q} T_{ij}\|v_i - o_j\|^2}_{\textit{Subspace mixture cost}} \geq \underbrace{\alpha_p \sigma_p^2 + \beta_q \sigma_q^2}_{\textit{Subspace specialization cost}} + \underbrace{\frac{\alpha_p \beta_q}{\alpha_p + \beta_q}\|\mu_p - \mu_q\|^2}_{\textit{Penalty}}. \quad (10)
$$

*Since $\|\mu_p - \mu_q\| > 0$ for any pair of distinct clusters, the inequality implies that mixing clusters in a single subspace always incurs a strictly higher OT cost. Equality holds only when $\mu_p = \mu_q$, i.e., $\mathcal{C}_p$ and $\mathcal{C}_q$ are the same cluster. Therefore, the OT minimizer naturally avoids mixing and induces subspace specialization. Details please refer to Appendix J.*

### 3.3 TRAINING AND INFERENCE

**Training** We train TokenCLIP in an end-to-end manner to capture global anomaly semantics $L_g$, local anomaly semantics $L_{\text{base}}$ and $L_{\text{da}}$. In addition to the regularization term $L_{\text{reg}}$, we introduce a hinge loss to explicitly enforce separation between normal regions and anomalous regions. Let the normal and anomaly indices be defined as: $\mathbb{I}_n = \{i \mid S(i) = 0\}$ and $\mathbb{I}_a = \{i \mid S(i) = 1\}$, where $S(i)$ denotes the ground-truth pixel-level annotation. The hinge loss is formulated as:

$$
L_{\text{hinge}} = \frac{1}{|\mathbb{I}_n|} \sum_{i \in \mathbb{I}_n} \max(S_n^{\text{da}}(i) - \delta^-, 0) + \frac{1}{|\mathbb{I}_a|} \sum_{i \in \mathbb{I}_a} \max(\delta^+ - S_a^{\text{da}}(i), 0),
$$

where $S_a^{\text{da}}(i)$ is the predicted anomaly score for token $i$, and $\delta^-, \delta^+$ are thresholds for enforcing margin constraints. The total loss is:

$$
L_{\text{total}} = L_g + L_{\text{base}} + L_{\text{da}} + \eta L_{\text{hinge}} + \xi L_{\text{reg}}, \quad (11)
$$

where $\eta, \xi$ are weighting coefficients for the global loss, regularization loss, base local loss, dynamic alignment loss, and hinge loss, respectively.

**Inference** Given an image $x$, TokenCLIP could simultaneously provides image-level anomaly score $\mathcal{A}_I(x)$ and pixel-level segmentation $\mathcal{A}_S(x)$. The pixel-level anomaly score combines anomaly segmentation from indiscriminate alignment and dynamic alignment $\mathcal{A}_S(x) = \frac{1}{2}(S_a^{da} + S_a)$. Considering the maximum anomaly score of local anomaly could reflect the image-level anomaly, the image-level anomaly score is given as $\mathcal{A}_I(x) = \frac{1}{2}(P_a(\bar{g}_a, f) + \frac{1}{2}\max(\mathcal{A}_S(x)))$.

Table 1: ZSAD performance on industrial domain datasets. Best: Red; Second-best: Blue.

| Task | Dataset | CoOp (IJCV'22) | WinCLIP (CVPR'23) | VAND (ARXIV'23) | AdaCLIP (ECCV'24) | AnomalyCLIP (ICLR'24) | FAprompt (ICCV'25) | TokenCLIP (Ours) |
|---|---|---|---|---|---|---|---|---|
| Image-level (AUROC, AP) | MVTec AD | (88.8, 94.8) | (91.8, 96.5)† | (86.1, 93.5)† | (89.6, -)† | (91.5, 96.2)† | (90.8, 94.9) | (93.5, 96.7) |
| | VisA | (62.8, 68.1) | (78.1, 81.2)† | (78.0, 81.4)† | (83.8, -)† | (82.1, 85.4)† | (83.6, 85.6) | (85.8, 88.2) |
| | MPDD | (55.1, 64.2) | (63.6, 69.9) | (73.0, 80.2) | (76.8, -)† | (77.0, 82.0)† | (77.5, 82.2) | (80.0, 82.3) |
| | BTAD | (66.8, 77.4) | (68.2, 70.9) | (73.6, 68.6) | (88.6, -)† | (88.3, 87.3)† | (92.3, 93.0) | (91.0, 90.5) |
| | SDD | (74.9, 65.1) | (84.3, 77.4) | (79.8, 71.4) | (-, -) | (84.7, 80.0)† | (81.8, 77.5) | (88.1, 85.2) |
| | DAGM | (87.5, 74.6) | (91.8, 79.5) | (94.4, 83.8) | (98.3, -)† | (97.5, 92.3)† | (96.9, 90.6) | (98.7, 95.2) |
| | DTD-Synthetic | (-, -) | (93.2, 92.6) | (86.4, 95.0) | (95.5, -)† | (93.5, 97.0)† | (95.6, 97.4) | (95.8, 97.6) |
| Pixel-level (AUROC, PRO) | MVTec AD | (33.3, 6.7) | (85.1, 64.6)† | (87.6, 44.0)† | (90.3, -)† | (91.1, 81.4)† | (90.6, 81.6) | (92.2, 87.9) |
| | VisA | (24.2, 3.8) | (79.6, 56.8)† | (94.2, 86.8)† | (95.6, -)† | (95.5, 87.0)† | (95.6, 86.7) | (95.9, 88.5) |
| | MPDD | (15.4, 2.3) | (76.4, 48.9) | (94.1, 83.2) | (96.4, -)† | (96.5, 88.7)† | (95.7, 85.6) | (96.8, 89.3) |
| | BTAD | (28.6, 3.8) | (72.7, 27.3) | (60.8, 25.0) | (92.1, -)† | (94.2, 74.8)† | (94.8, 75.1) | (95.1, 78.3) |
| | SDD | (28.9, 7.1) | (68.8, 24.2) | (79.8, 65.1) | (-, -) | (90.6, 67.8)† | (92.5, 70.0) | (90.8, 70.3) |
| | DAGM | (17.5, 2.1) | (87.6, 65.7) | (82.4, 66.2) | (91.0, -)† | (95.6, 91.0)† | (98.1, 94.9) | (95.8, 91.6) |
| | DTD-Synthetic | (-, -) | (83.9, 57.8) | (95.3, 86.9) | (96.9, -)† | (97.9, 92.3)† | (98.0, 92.2) | (98.1, 93.7) |

## 4 EXPERIMENTS

**Dataset Details & Baselines**  We evaluate TokenCLIP in ZSAD through large-scale experiments. The evaluation covers two distinct domains: industrial inspection and medical diagnosis. In the industrial domain, we evaluate seven benchmarks: MVTec AD (Bergmann et al., 2019), VisA (Zou et al., 2022), MPDD (Jezek et al., 2021), BTAD (Mishra et al., 2021), SDD (Tabernik et al., 2020), DAGM (Wieler & Hahn, 2007), and DTD-Synthetic (Aota et al., 2023). These datasets span various manufactured objects and defect types. In the medical domain, we assess tasks including skin lesion detection (ISIC (Gutman et al., 2016)), colon polyp segmentation (CVC-ClinicDB (Bernal et al., 2015), CVC-ColonDB (Tajbakhsh et al., 2015), Kvasir (Jha et al., 2020), Endo (Hicks et al., 2021)), and brain anomaly detection (HeadCT, BrainMRI (Salehi et al., 2021), Br35H (Hamada., 2020)). Detailed dataset, implementation, and evaluation metric are provided in Appendices A, B, and C.

**Implementation details**  We use the publicly available CLIP model (ViT-L/14@336px) as the backbone. For data preprocessing, we adopt the same pipeline as AnomalyCLIP to ensure fair comparison. All input images are resized to $518 \times 518$. We use the top feature as the set of visual patch tokens $\{v_i\}_{i=1}^{N}$. The length of the learnable text embedding is set to $E = 12$. For the textual subspace configuration, we use $Q = 3$ subspaces when training on the VisA dataset, and $Q = 4$ subspaces for MVTec AD. The OT problem is solved using the Sinkhorn-Knopp algorithm with 100 iterations, and the entropic regularization coefficient $\lambda$ is set to 0.01. The marginal vectors $\mathbf{u}$ and $\mathbf{v}$ are initialized as uniform probability distributions. Accordingly, their weights are set to : $p_i = \frac{1}{N}$ and $q_j = \frac{1}{Q}$. We set the sparsification threshold parameter $\epsilon = 0.2$ and retain the top 2 entries per row in the transport plan. The loss weights $\eta, \xi$ are set to 5 and 100, respectively. We use Adama optimizer with a learning rate of 1e-3 with batch size 8. The training epoch is 30. All experiments are conducted using PyTorch 2.0.0. † denotes results taken from original papers.

### 4.1 MAIN RESULTS

**ZSAD performance on industrial defect detection**  We evaluate the effectiveness of TokenCLIP on ZSAD task across seven industrial datasets spanning diverse object categories. As presented in Table 1, TokenCLIP outperforms state-of-the-art baselines. On MVTec AD, it achieves a pixel-level performance of 92.2 AUROC and 87.9 PRO, surpassing AnomalyCLIP's 91.1 AUROC and 81.4 PRO. Notably, TokenCLIP demonstrates significant improvements in PRO, underscoring its superiority in detecting fine-grained and subtle anomalies. These improvements are primarily attributed to its dynamic token-level textual supervision, which enables more precise and semantics-aware alignment for anomaly modeling. In addition to pixel-level gains, TokenCLIP also shows obvious improvements in image-level anomaly detection. It stems from the decoupling modeling of global and local anomaly semantics. This shows that TokenCLIP captures fine-grained, generalized anomaly semantics. We observe that FAPrompt is a competitive model for pixel-level segmentation. However, it requires higher computational overhead to learn multiple learnable prompts.

**ZSAD performance on cross-domain Medical analysis**  To further demonstrate the advantage of dynamic alignment over indiscriminate alignment, we use the checkpoint trained on MVTec AD to evaluate performance on medical datasets directly. As shown in Table 2, TokenCLIP consistently

Table 2: Cross-domain ZSAD performance on medical analysis. Best: Red; Second-best: Blue.

| Task | Dataset | CoOp (IJCV'22) | WinCLIP (CVPR'23) | VAND (ARXIV'23) | AdaCLIP (ECCV'24) | AnomalyCLIP (ICLR'24) | FAprompt (ICCV'25) | TokenCLIP (Ours) |
|---|---|---|---|---|---|---|---|---|
| Image-level (AUROC, AP) | HeadCT | (78.4, 78.8) | (81.8, 80.2) | (89.1, 89.4) | (91.5, -) | (93.4, 91.6) | (93.9, 92.6) | (96.0, 95.3) |
| | BrainMRI | (61.3, 44.9) | (86.6, 91.5) | (89.3, 90.9) | (94.8, -) | (90.3, 92.2) | (94.8, 93.7) | (95.3, 95.8) |
| | Br35H | (86.0, 87.5) | (80.5, 82.2) | (93.1, 92.9) | (97.7, -) | (94.6, 94.7) | (96.6, 95.6) | (97.8, 97.6) |
| Pixel-level (AUROC, PRO) | ISIC | (51.7, 15.9) | (83.3, 55.1) | (89.4, 77.2) | (88.3, -) | (89.7, 78.4) | (90.7, 80.3) | (91.6, 83.4) |
| | ColonDB | (40.5, 2.60) | (70.3, 32.5) | (78.4, 64.6) | (79.1, -) | (81.9, 71.3) | (84.1, 73.2) | (83.8, 72.8) |
| | ClinicDB | (34.8, 2.40) | (51.2, 13.8) | (80.5, 60.7) | (84.4, -) | (82.9, 67.8) | (83.9, 69.3) | (84.2, 69.7) |
| | Kvasir | (44.1, 3.50) | (69.7, 24.5) | (75.0, 36.2) | (-, -) | (78.9, 45.6) | (80.4, 46.5) | (80.6, 46.9) |
| | Endo | (40.6, 3.90) | (68.2, 28.3) | (81.9, 54.9) | (-, -) | (84.1, 63.6) | (85.9, 65.0) | (86.1, 66.2) |

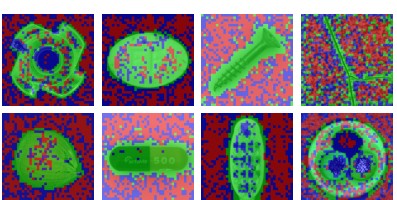

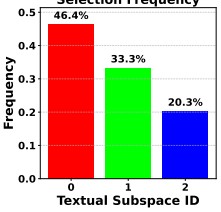

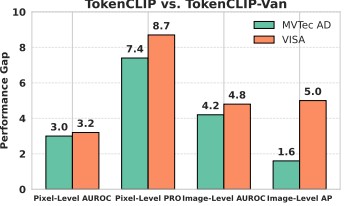

(a) Token-level assigned textual subspace. (b) Selection frequency. (c) TokenCLIP vs. TokenCLIP-Van.

Figure 3: (a) Visualization of the assigned textual subspaces for patch token; Different color denotes different textual subspaces. (b) Selection frequency of textual subspaces across MVTec AD, where the color corresponds to that in (a). (c) Performance gap between TokenCLIP and TokenCLIP-Van.

outperforms other methods at both the image and pixel levels. On ISIC, TokenCLIP achieves 91.6 AUROC and 83.4 PRO, compared to the second-best performance of 90.7 AUROC and 80.3 PRO. n addition, image-level performance on HeadCT, BrainMRI, and Br35H demonstrates substantial improvements, further confirming the model's ability to capture generalized anomaly semantics.

## 4.2 RESULT ANALYSIS

**Fine-grained alignment for token-level modeling** In this section, we analyze the semantic patterns captured by the textual subspaces during dynamic alignment. As shown in Figure 3(a), the green textual subspace is frequently assigned to foreground regions of objects such as nuts, pills, screws, and tiles. This indicates that it captures object-centric semantics. In contrast, the red and blue textual subspaces are predominantly distributed across background regions. It suggests that they primarily model contextual or low-variation areas. Furthermore, the green subspace tends to concentrate in regions with significant semantic variation, while the red and blue subspaces are more commonly associated with smooth or homogeneous textures. For example, in the tile image, the surface is largely uniform, but the crack introduces a distinct visual change, precisely where the green subspace becomes dominant. In cable, the missing wire appears as a flat black region with low-variance semantic nature and is typically assigned to the red or blue subspaces. We can conclude that the learned textual subspaces serve distinct semantic roles: one subspace captures object-level and variant semantics, while the others are more aligned with background and uniform regions. In addition, we analyze the selection frequency of each textual subspace. Figure 3(b) presents the frequency distribution of the selected textual subspaces across the MVTec AD dataset. We observe that the blue and green subspaces account for the majority of selections. This indicates that background regions or areas with low semantic variation occupy a large portion of the images.

**OT is important to dynamic alignment** We investigate the role of OT in dynamic alignment by comparing TokenCLIP with a variant called TokenCLIP-Van, which replaces OT with a simpler mechanism that directly selects the textual subspace that has the highest cosine similarity. All subsequent steps, including top-$k$ sparsification and row-wise normalization, remain unchanged. As shown in Figure 3(c), TokenCLIP consistently outperforms TokenCLIP-Van across all evaluation metrics. This

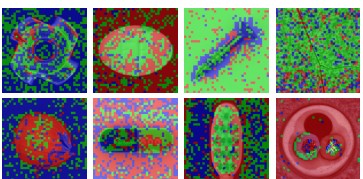

Figure 4: Token-level assignment.

improvement arises from the marginal constraints and minimal cost objective of the OT formulation, which globally optimizes the assignment between visual patch tokens and textual subspaces, allowing textual subspaces sufficient optimization and semantics specialization. In contrast, TokenCLIP-Van relies solely on local cosine similarity. Without global regularization, as illustrated in Figure 4,

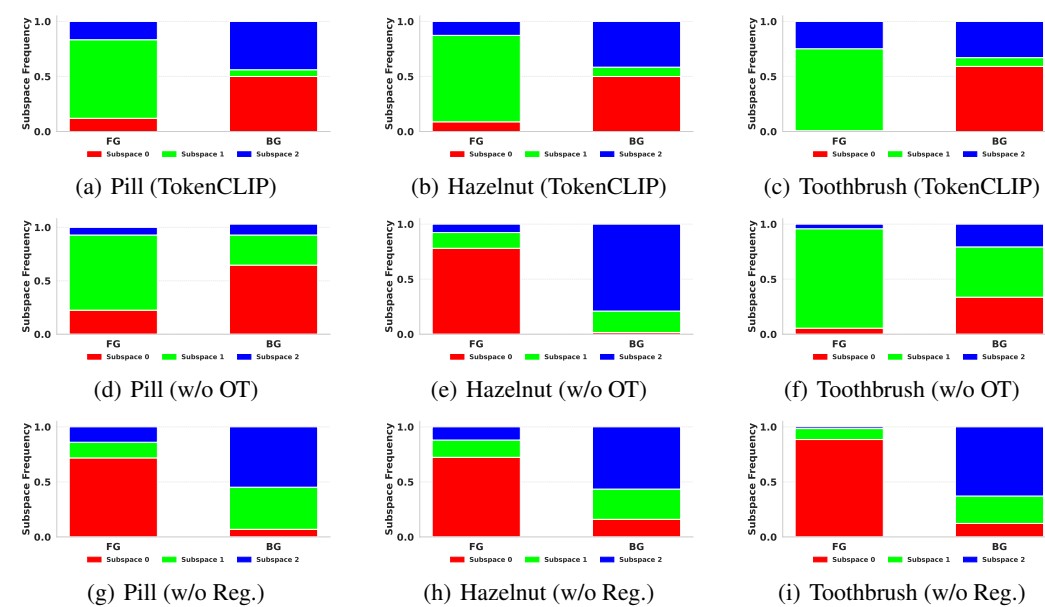

(a) Pill (TokenCLIP)    (b) Hazelnut (TokenCLIP)    (c) Toothbrush (TokenCLIP)

(d) Pill (w/o OT)    (e) Hazelnut (w/o OT)    (f) Toothbrush (w/o OT)

(g) Pill (w/o Reg.)    (h) Hazelnut (w/o Reg.)    (i) Toothbrush (w/o Reg.)

Figure 5: Subspace assignment frequency on foreground (FG) and background (BG) across different variants: Top: TokenCLIP, Middle: without OT, Bottom: without orthogonality regularization.

textual subspaces fail to collectively and distinctly capture different semantics, since each feature only greedily matches more visual tokens during training. When one subspace dominates the matching, others receive limited optimization and struggle to achieve accurate alignment. Consequently, although TokenCLIP-Van ensembles multiple text prompts, it fails to capture fine-grained semantics and sometimes even performs worse than indiscriminate alignment. These findings underscore the importance of OT in enabling precise and fine-grained alignment for generalized anomaly modeling.

**What makes the textual specialization** Here, we discuss the contributions of OT and orthogonality regularization to textual specialization. To this end, we visualize the subspace assignment frequency on background and foreground regions across different categories. The first row shows the results of TokenCLIP. The green subspace mainly focuses on object foregrounds (FG), while the blue and red subspaces focus on backgrounds (BG). This demonstrates that TokenCLIP success-

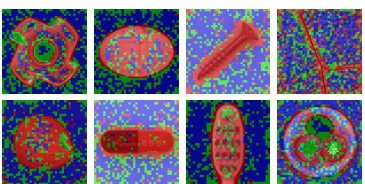

Figure 6: Assignment w/o Reg.

fully achieves subspace specialization, consistent with the observation in Figure 3(a). The middle row shows TokenCLIP without OT. In this case, no clear textual specialization is observed. For example, in the Toothbrush category, the green subspace dominates both FG and BG assignments, while in Hazelnut it is rarely used. This indicates that OT plays a dominant role in enabling fine-grained semantic specialization. The bottom row shows TokenCLIP without orthogonality regularization. The red subspace dominates the FG regions, while the blue and green subspaces are mainly assigned to the BG regions. The specialization still emerges even without orthogonality regularization, but the extent of specialization is weaker than that of the full TokenCLIP model. We additionally visualize the token-level assignments of TokenCLIP without orthogonality regularization in Figure 6.

**Computation overhead** Token-CLIP efficiently provides token-level supervision by combining a limited number of textual subspaces, rather than assigning a dedicated

Table 3: Analysis of computation overhead.

| Methods | Training per epoch | Inference Time | FPS | Peak GPU Memory (MB) | MVTec AD pixel-level | VisA pixel-level |
|---|---|---|---|---|---|---|
| AnomalyCLIP | 96s | 0.124s | 8.04 | 2235MB | (91.1, 81,4) | (95.5, 87.0) |
| FAprompt | 245s | 0.214s | 4.67 | 4238MB | (90.6, 81,6) | (95.6, 86.7) |
| TokenCLIP | 110s | 0.135s | 7.39 | 3186MB | (92.2, 87.9) | (95.9, 88.5) |

textual space to each visual token. We measure GPU memory consumption during training and report inference speed in terms of frames per second (FPS). All experiments are conducted on an idle GPU with a batch size of 1. Table 3 shows that TokenCLIP incurs only a slight increase in inference time and GPU memory usage compared to AnomalyCLIP. In contrast, FAPrompt requires 1.9× more GPU memory and nearly double the inference time. As for performance, TokenCLIP outperforms FAPrompt and strikes a good balance between performance and computational overhead.

Table 4: Module Ablation.

| Module | | | | MVTec AD | | VisA | |
|---|---|---|---|---|---|---|---|
| $T_1$ | $T_2$ | $T_3$ | $T_4$ | Pixel-level | Image-level | Pixel-level | Image-level |
| | | | | (80.3, 77.8) | (89.9, 95.4) | (86.6, 78.1) | (82.2, 84.9) |
| ✓ | | | | (91.1, 81,4) | (91.5, 96.2) | (95.5, 87.0) | (82.1, 85.4) |
| ✓ | ✓ | | | (91.8, 87.0) | (93.1, 96.6) | (95.6, 87.8) | (84.5, 88.0) |
| ✓ | ✓ | ✓ | | (91.7, 87.4) | (93.2, 96.8) | (95.8, 88.2) | (85.1, 87.9) |
| ✓ | ✓ | | ✓ | (91.8, 87.2) | (92.9, 96.0) | (95.7, 88.0) | (85.2, 87.5) |
| ✓ | | ✓ | ✓ | (91.5, 83.2) | (92.4, 96.3) | (95.2, 87.6) | (83.0, 85.8) |
| | ✓ | ✓ | ✓ | (83.2, 62.5) | (89.6, 94.8) | (95.2, 87.1) | (84.8, 86.7) |
| ✓ | ✓ | ✓ | ✓ | (92.2, 87.9) | (93.5, 96.7) | (95.9, 88.5) | (85.8, 88.2) |

Table 5: Subspace number ablation.

| Subspace number $Q$ | MVTec AD | | VisA | |
|---|---|---|---|---|
| | Pixel-level | Image-level | Pixel-level | Image-level |
| 1 | (91.6, 85.3) | (92.1, 96.2) | (95.3, 87.2) | (83.6, 85.9) |
| 2 | (91.8, 86.8) | (93.0, 96.2) | (95.5, 87.4) | (84.2, 86.8) |
| 3 | (92.2, 87.9) | (93.5, 96.7) | (95.7, 87.8) | (85.3, 87.7) |
| 4 | (91.8, 86.8) | (93.2, 96.5) | (95.9, 88.5) | (85.8, 88.2) |
| 5 | (91.5, 86.2) | (93.3, 96.1) | (95.1, 87.2) | (85.0, 87.8) |

## 5 ABLATION STUDY

**Module Ablation**   In this section, we investigate the contributions of four key modules to the overall performance of TokenCLIP: base semantics learning ($T_1$), OT ($T_2$), decoupled local and global text prompts ($T_3$), and orthogonal regularization ($T_4$). The vanilla model, which includes only $T_1$, captures global anomaly semantics and serves as the basic baseline. Adding base local anomaly modeling ($T_2$) yields a notable performance boost, corresponding to AnomalyCLIP. This gain is primarily attributed to the indiscriminate alignment that enables learning a token-agnostic unified textual space. Decoupling global and local anomaly semantics ($T_3$) further improves both image-level and pixel-level performance by allowing each prompt to specialize. Building on this, we incorporate orthogonal regularization ($T_4$) to encourage diversity among the textual subspaces. Additional gains are observed due to finer semantic specialization. Notably, when we remove $T_1$, TokenCLIP exhibits a significant performance drop on MVTec AD, while the drop is less pronounced on VisA. This suggests that base anomaly semantics serve as a foundation for learning and help stabilize and enhance the optimization of orthogonal subspaces.

**Number Ablation of Textual Subspaces.**   Textual subspaces enable TokenCLIP to perform fine-grained alignment by capturing diverse visual semantics. We conduct an ablation study to investigate how the number of subspaces influences anomaly detection performance. As shown in Table 5, using three and four subspaces yields the best overall results on MVTec AD and VisA, respectively. When the number of subspaces is too small (e.g., 1), the model fails to capture sufficient anomaly-related semantics. Increasing the number of subspaces from 1 to 3 leads to notable performance gains. However, using too many subspaces (e.g., 5) may result in suboptimal optimization due to fragmented semantic representations. An appropriate number of subspaces could promote TokenCLIP.

**K Ablation of topK**   The parameter $k$ determines the number of selected textual subspaces assigned to each visual patch token in the dynamic alignment process. In this section, we investigate how $k$ affects the performance of TokenCLIP. As shown in Table 6, increasing $k$ from 1 to 2 yields a noticeable perfor-

Table 6: Selected k ablation.

| topK | MVTec AD | | VisA | |
|---|---|---|---|---|
| | Pixel-level | Image-level | Pixel-level | Image-level |
| 1 | (91.7, 86.5) | (92.8, 96.3) | (95.6, 86.2) | (83.9, 85.3) |
| 2 | (92.2, 87.9) | (93.5, 96.7) | (95.9, 88.5) | (85.8, 88.2) |
| 3 | (91.8, 87.2) | (93.1, 96.5) | (95.7, 87.8) | (85.0, 87.5) |

mance improvement. Incorporating adequate textual subspace facilitates the capture of fine-grained anomaly semantics. However, further increasing $k$ from 2 to 3 introduces subspaces with unrelated semantics, which weakens the effectiveness of dynamic alignment. Therefore, TokenCLIP benefits most from an appropriate k that supports subspace specialization while avoiding semantic over-coupling.

## 6 CONCLUSION

This paper reveals that existing CLIP-based methods are limited by indiscriminate alignment, where all visual patch tokens are supervised using a single, token-agnostic textual space. To overcome this limitation, we propose a dynamic alignment mechanism that provides token-level supervision by adaptively assigning each patch token to a semantically-aware combination of textual subspaces. We formulate this assignment as an OT problem to encourage each subspace to specialize in distinct semantic patterns and to be sufficiently optimized. Extensive experiments demonstrate the effectiveness of TokenCLIP across multiple benchmarks.

**Limitations**   The dynamic alignment process introduces additional computational overhead. However, this minor increase is justified by the substantial improvement in overall anomaly detection performance. We provide analyses of failure cases in Appendix H.

**Broader Impacts**   Our project aims to improve the intelligence level of industrial monitoring systems. Our study on replacing indiscriminate alignment with dynamic alignment can improve the detection level of smart manufacturing. This research does not involve any violations of legal or ethical standards. We hope our work will inspire further research and development of ZSAD.

## REPRODUCIBILITY STATEMENT

We provide the dataset and baseline details in Appendices A and B. Ablation studies on the hinge loss coefficient $\xi$ and regularization loss coefficient $\eta$, as well as the threshold $\epsilon$, are included in Appendix D. We provide more visualization in Appendix E. We analyze the relation between subspace orthogonality and specialization in Appendix F. Appendix G presents the difference between subspaces with randomly initialized learnable vectors. To offer more insight into TokenCLIP, we present a failure case in Appendix H. We report category-wise results to facilitate fine-grained comparisons in Appendix I. Appendix J presents the detailed theoretical analysis. The code will be made available once accepted.

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

## A   DATASETS

This section provides a statistical overview of the 15 datasets utilized in our study, spanning both industrial and medical domains. Detailed descriptions are presented in Table 7.

Table 7: Overview of datasets used for anomaly and disease detection across industrial and medical domains.

| Domain | Dataset | Modality | $|\mathcal{C}|$ | Normal / Anomalous | Application |
|---|---|---|---|---|---|
| Industrial Inspection | MVTec AD | RGB | 15 | (467, 1258) | Defect detection |
| | VisA | RGB | 12 | (962, 1200) | Defect detection |
| | MPDD | RGB | 6 | (176, 282) | Defect detection |
| | BTAD | RGB | 3 | (451, 290) | Defect detection |
| | SDD | RGB | 1 | (181, 74) | Defect detection |
| | DAGM | RGB | 10 | (6996, 1054) | Defect detection |
| | DTD-Synthetic | RGB | 12 | (357, 947) | Defect detection |
| Skin Lesion Analysis | ISIC | RGB | 1 | (0, 379) | Skin cancer detection |
| Colon Polyp Detection | ClinicDB | Endoscopy | 1 | (0, 612) | Polyp detection |
| | ColonDB | Endoscopy | 1 | (0, 380) | Polyp detection |
| | Kvasir | Endoscopy | 1 | (0, 1000) | Polyp detection |
| | Endo | Endoscopy | 1 | (0, 200) | Polyp detection |
| Brain Tumor Detection | HeadCT | CT | 1 | (100, 100) | Tumor detection |
| | BrainMRI | MRI | 1 | (98, 155) | Tumor detection |
| | Br35H | MRI | 1 | (1500, 1500) | Tumor detection |

## B   BASELINES

Our approach replaces conventional static alignment with a dynamic alignment strategy and can serve as a plug-in for CLIP-based anomaly detection methods. Given the rapid evolution of the field, we compare TokenCLIP against several representative methods that adopt static alignment: CoOp (Zhou et al., 2022), WinCLIP (Jeong et al., 2023), VAND (Chen et al., 2023b), AnomalyCLIP (Zhou et al., 2024a), AdaCLIP (Cao et al., 2024), and FAprompt (ZHU et al., 2025).

- CoOp (IJCV 2022) (Zhou et al., 2022): A prompt optimization method that replaces fixed text templates with learnable embeddings. Following Zhou et al. (2024a), we construct prompts by inserting tokens representing normal or anomalous conditions before the class name. Specifically, the templates take the form $[V_1][V_2]...[V_N]$ `normal` `[cls]` and $[V_1][V_2]...[V_N]$ `anomalous` `[cls]`.

- WinCLIP (CVPR 2023) (Jeong et al., 2023): They leverage a comprehensive set of handcrafted prompts tailored to anomaly scenarios. It introduces a window-based scaling mechanism to enhance anomaly localization. We reproduce the experimental setup as reported in the original publication for consistency.

- VAND (ARXIV 2023) (Chen et al., 2023b): This method advances prompt design by incorporating learnable linear projections, allowing it to better capture fine-grained visual semantics. We adopt the original implementation and settings to align with the authors' reported results.

- AdaCLIP (ECCV 2024) (Cao et al., 2024): AdaCLIP retains handcrafted textual prompts but focuses on adapting the visual embedding space to improve anomaly detection.

- AnomalyCLIP (ICLR 2024) (Zhou et al., 2022): AnomalyCLIP introduces object-agnostic prompt learning and achieves promising generalization across different types of anomalies. It also adapts the textual and visual spaces for better detection performance.

- FAPromt (ICCV 2025) (ZHU et al., 2025): FAPromt proposes to use multiple learnable text prompts to learn complementary and decomposed abnormality. prompts

## C EVALUATION SETTING AND METRICS

We evaluate image-level detection using AUROC and Average Precision (AP). For pixel-level segmentation, we report AUROC and AUPRO. While AUROC reflects overall pixel-wise discrimination, AUPRO emphasizes the quality of region-level anomaly localization. All results are averaged over five independent runs for robustness. Following the evaluation setting (Zhou et al., 2024a), MVTec AD serves as the auxiliary training set when testing on other datasets. Conversely, VisA is used for training when evaluating on MVTec AD. Final results are computed by averaging over all sub-datasets within each benchmark.

## D HYPERPARAMETER ABLATION

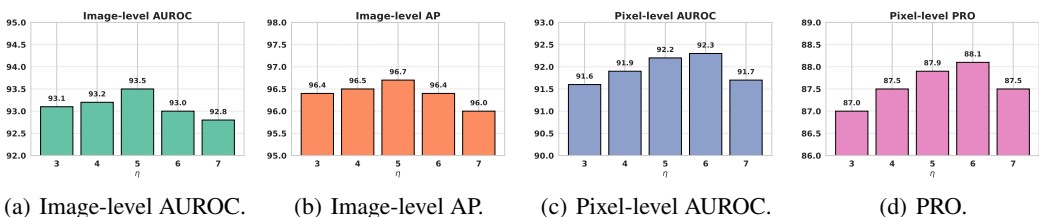

(a) Image-level AUROC.    (b) Image-level AP.    (c) Pixel-level AUROC.    (d) PRO.

Figure 7: The hinge loss effect of $\eta$

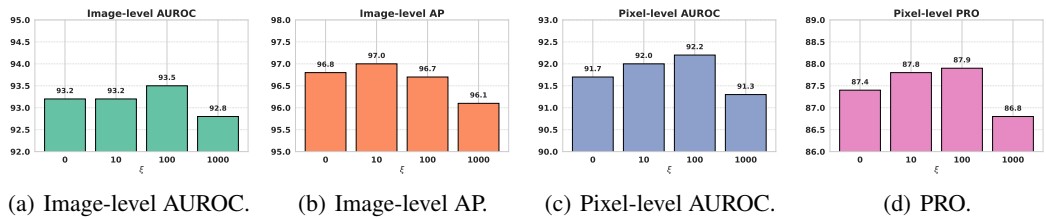

(a) Image-level AUROC.    (b) Image-level AP.    (c) Pixel-level AUROC.    (d) PRO.

Figure 8: The regularization loss effect of $\xi$

Here, we investigate the effect of the loss coefficients, i.e., $\eta$ and $\xi$. As shown in Figure 7, we observe that setting $\eta = 5$ leads to improvements in both image-level and pixel-level performance. However, as $\eta$ continues to increase, the performance begins to decline. This shows that excessive regularization can hinder the learning of the main objective. A similar phenomenon can also be observed for $\xi$.

**Threshold Ablation** When a textual subspace closely matches a visual patch token, the top-1 subspace typically receives a dominant weight, while the remaining $k - 1$ subspaces contribute less. Here, we perform an ablation study to evaluate the effect of thresholding low-weight assignments. As shown in Table 8, a threshold of 0.2 yields the best overall per-

Table 8: Threshold value ablation.

| $\epsilon$ | MVTec AD | | VisA | |
|---|---|---|---|---|
| | Pixel-level | Image-level | Pixel-level | Image-level |
| 0.1 | (92.1, 87.7) | (93.5, 97.0) | (95.6, 88.0) | (85.1, 87.7) |
| 0.2 | (92.2, 87.9) | (93.5, 96.7) | (95.9, 88.5) | (85.8, 88.2) |
| 0.3 | (92.0, 87.2) | (93.3, 96.5) | (95.6, 87.3) | (84.6, 87.1) |
| 0.4 | (91.8, 86.7) | (93.0, 96.5) | (95.6, 86.5) | (84.2, 85.8) |

formance across both MVTec AD and VisA. This suggests that a moderate threshold can effectively eliminate relatively irrelevant subspaces, thereby promoting subspace specialization. However, higher thresholds (e.g., 0.3 or 0.4) remove semantically relevant subspaces and impair accurate alignment. Selecting an appropriate threshold is therefore critical to balancing subspace specialization and semantic comprehensiveness.

## E VISUALIZATION

We provide a visualization comparison between TokenCLIP and TokenCLIP-Van to offer an intuitive illustration of their differences. As shown in Figure 9, TokenCLIP exhibits more robust performance than TokenCLIP-Van across both industrial and medical domains.

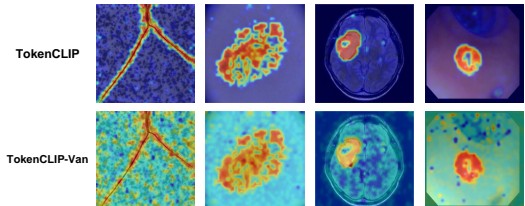

Figure 9: Visualization comparison.

## F ANALYSIS ON SUBSPACE TEXTUAL ORTHOGONALITY AND SPECIALIZATION

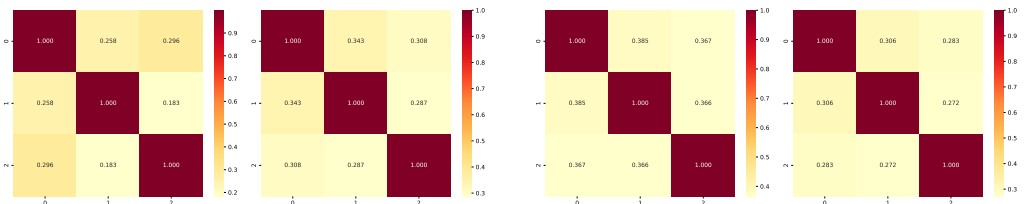

(a) Subspace orthogonality of TokenCLIP. Left: textual normality subspaces; Right: textual abnormality subspaces.

(b) Subspace orthogonality of TokenCLIP w/o OT. Left: textual normality subspaces; Right: textual abnormality subspaces.

Figure 10: Visualization of subspace textual orthogonality.

We clarify that the purpose of introducing orthogonality regularization is to prevent the specialized textual subspaces from excessively overlapping, rather than to create the specialization itself. The textual specialization in TokenCLIP is primarily induced by OT. Importantly, good subspace specialization does not necessarily imply that the subspaces should become more orthogonal in the embedding space. Each subspace is expected to align with a different visual semantic pattern. Since these visual semantics are inherently correlated, the corresponding textual subspaces are not required to be strictly orthogonal. Instead, sufficient separation is enough for effective specialization.

To verify this, we compute the cosine similarity between the learned textual subspaces for both the OT-enabled model and the variant without OT. Note that values close to 1 correspond to near-parallel vectors, whereas values close to 0 correspond to near-orthogonal vectors. As shown in Figure 10, abnormality textual subspaces in TokenCLIP with OT are less orthogonal than those without OT. However, the latter is inferior in learning specialized textual semantics, as analyzed in Figure 5.

## G DIFFERENCE BETWEEN SUBSPACES WITH RANDOMLY INITIALIZED LEARNABLE VECTORS

We fine-grain the original textual space into multiple subspaces to specialize different semantics because CLIP text prompts are originally designed to encode generalized semantic knowledge. An interesting question is whether TokenCLIP would still work if we replaced the textual subspaces with randomly initialized, learnable vectors. To examine this, we remove the text encoder and initialize the subspaces as learnable random vectors while keeping the rest of the framework unchanged. We refer to this variant as TokenCLIP-Visual. On MVTec AD, TokenCLIP-Visual obtains pixel-level performance of 84.5 AUROC and 82.3 PRO, which are inferior to the full TokenCLIP model, achieving 92.2 AUROC and 87.9 PRO. We attribute this performance degradation to single-modal alignment: TokenCLIP-Visual eliminates the text encoder entirely, and its learnable subspaces effectively act as clusters over the visual semantics extracted from the auxiliary dataset. The alignment is performed purely within a visual–visual embedding space. However, the strong zero-shot capability of CLIP originates from its joint visual–text joint space, learned through large-scale contrastive pretraining. TokenCLIP-Visual loses this cross-modal alignment, which is crucial for generalizing to unseen test

categories in a zero-shot setting. In contrast, TokenCLIP fine-grains the textual space and inherits the strong cross-modality zero-shot generalization.

## H FAILURE CASE

We further analyze the behavior of TokenCLIP by examining its failure cases. Figure 11(a) presents examples of false positives in segmentation results. As highlighted by the yellow circles in Figure 11(b)(a), TokenCLIP occasionally produces spot-level false alarms. To investigate the cause, we visualize the corresponding token-level textual subspace assignments in Figure 11(a)(b). We observe that these false positives often coincide with inconsistencies in subspace assignment within a localized region. For example, in the capsule image, the top yellow circle corresponds to a single token assigned to the blue subspace amid a predominantly green region. A similar pattern appears in the tile image, where the false detection at the top aligns with an abrupt shift in subspace assignment. These observations suggest that spatial inconsistency in textual subspace alignment may lead to local false positives.

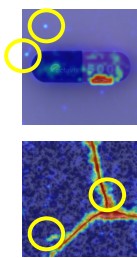 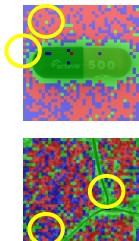

(a) Segmentation visualization.          (b) Corresponding textual subspace assignment.

Figure 11: Visualization of false detections. (a) Segmentation outputs from TokenCLIP with false positives highlighted in yellow. (b) Corresponding token-level textual subspace assignments, where inconsistent assignments correlate with the observed false positives.

## I SUBSET-LEVEL RESULTS

To provide a more detailed evaluation, we report the subset–level performance in the following tables.

Table 9: Subset-level performance comparison (AUROC) for anomaly segmentation on MVTec AD.

| Object name | WinCLIP | VAND | CoOp | AnomalyCLIP | TokenCLIP |
|---|---|---|---|---|---|
| Carpet | 95.4 | 98.4 | 6.7 | 98.8 | 99.0 |
| Bottle | 89.5 | 83.4 | 23.1 | 90.4 | 91.7 |
| Hazelnut | 94.3 | 96.1 | 30.2 | 97.1 | 97.6 |
| Leather | 96.7 | 99.1 | 11.7 | 98.6 | 99.4 |
| Cable | 77.0 | 72.3 | 49.7 | 78.9 | 81.4 |
| Capsule | 86.9 | 92.0 | 35.5 | 95.8 | 96.9 |
| Grid | 82.2 | 95.8 | 7.8 | 97.3 | 98.5 |
| Pill | 80.0 | 76.2 | 46.5 | 92.0 | 92.5 |
| Transistor | 74.7 | 62.4 | 50.1 | 71.0 | 69.8 |
| Metal_nut | 61.0 | 65.4 | 49.3 | 74.4 | 74.5 |
| Screw | 89.6 | 97.8 | 17.0 | 97.5 | 98.3 |
| Toothbrush | 86.9 | 95.8 | 64.9 | 91.9 | 94.8 |
| Zipper | 91.6 | 91.1 | 33.4 | 91.4 | 96.5 |
| Tile | 77.6 | 92.7 | 41.7 | 94.6 | 96.0 |
| Wood | 93.4 | 95.8 | 31.4 | 96.5 | 97.5 |
| Mean | 85.1 | 87.6 | 33.3 | 91.1 | 92.2 |

Table 10: Subset-level performance comparison (PRO) for anomaly segmentation on MVTec AD.

| Object name | WinCLIP | VAND | CoOp | AnomalyCLIP | TokenCLIP |
|---|---|---|---|---|---|
| Carpet | 84.1 | 48.5 | 0.50 | 90.1 | 97.8 |
| Bottle | 76.4 | 45.6 | 4.50 | 80.9 | 83.6 |
| Hazelnut | 81.6 | 70.3 | 4.70 | 92.4 | 92.5 |
| Leather | 91.1 | 72.4 | 1.80 | 92.2 | 98.2 |
| Cable | 42.9 | 25.7 | 12.2 | 64.4 | 74.1 |
| Capsule | 62.1 | 51.3 | 5.70 | 87.2 | 95.2 |
| Grid | 57.0 | 31.6 | 1.00 | 75.6 | 94.0 |
| Pill | 65.0 | 65.4 | 3.20 | 88.2 | 94.6 |
| Transistor | 43.4 | 21.3 | 9.30 | 58.1 | 56.6 |
| Metal_nut | 31.8 | 38.4 | 7.00 | 71.0 | 74.3 |
| Screw | 68.5 | 67.1 | 6.40 | 88.0 | 92.2 |
| Toothbrush | 67.7 | 54.5 | 16.6 | 88.5 | 91.2 |
| Zipper | 71.7 | 10.7 | 11.6 | 65.3 | 88.2 |
| Tile | 51.2 | 26.7 | 10.1 | 87.6 | 91.6 |
| Wood | 74.1 | 31.1 | 5.10 | 91.2 | 95.5 |
| Mean | 64.6 | 44.0 | 6.70 | 81.4 | 87.9 |

Table 11: Subset-level performance comparison (AUROC) for anomaly classification on MVTec AD.

| Object name | WinCLIP | VAND | CoOp | AnomalyCLIP | TokenCLIP |
|---|---|---|---|---|---|
| Carpet | 100.0 | 99.5 | 99.9 | 100.0 | 100.0 |
| Bottle | 99.2 | 92.0 | 87.7 | 89.3 | 92.2 |
| Hazelnut | 93.9 | 89.6 | 93.5 | 97.2 | 93.1 |
| Leather | 100.0 | 99.7 | 99.9 | 99.8 | 100.0 |
| Cable | 86.5 | 88.4 | 56.7 | 69.8 | 85.3 |
| Capsule | 72.9 | 79.9 | 81.1 | 89.9 | 95.4 |
| Grid | 98.8 | 86.3 | 94.7 | 97.0 | 99.0 |
| Pill | 79.1 | 80.5 | 78.6 | 81.8 | 91.8 |
| Transistor | 88.0 | 80.8 | 92.2 | 92.8 | 90.4 |
| Metal_nut | 97.1 | 68.4 | 85.3 | 93.6 | 88.8 |
| Screw | 83.3 | 84.9 | 88.9 | 81.1 | 83.8 |
| Toothbrush | 88.0 | 53.8 | 77.5 | 84.7 | 88.4 |
| Zipper | 91.5 | 89.6 | 98.8 | 98.5 | 97.9 |
| Tile | 100.0 | 99.9 | 99.7 | 100.0 | 99.4 |
| Wood | 99.4 | 99.0 | 97.7 | 96.8 | 98.6 |
| Mean | 91.8 | 86.1 | 88.8 | 91.5 | 93.5 |

Table 12: Subset-level performance comparison (AP) for anomaly classification on MVTec AD.

| Object name | WinCLIP | VAND | CoOp | AnomalyCLIP | TokenCLIP |
|---|---|---|---|---|---|
| Carpet | 100.0 | 99.8 | 100.0 | 100.0 | 99.7 |
| Bottle | 99.8 | 97.7 | 96.4 | 97.0 | 97.8 |
| Hazelnut | 96.9 | 94.8 | 96.7 | 98.6 | 96.6 |
| Leather | 100.0 | 99.9 | 100.0 | 99.9 | 99.9 |
| Cable | 91.2 | 93.1 | 69.4 | 81.4 | 89.8 |
| Capsule | 91.5 | 95.5 | 95.7 | 97.9 | 98.9 |
| Grid | 99.6 | 94.9 | 98.1 | 99.1 | 99.4 |
| Pill | 95.7 | 96.0 | 94.2 | 95.4 | 98.3 |
| Transistor | 87.1 | 77.5 | 90.2 | 90.6 | 88.9 |
| Metal_nut | 99.3 | 91.9 | 96.3 | 98.5 | 97.5 |
| Screw | 93.1 | 93.6 | 96.2 | 92.5 | 93.7 |
| Toothbrush | 95.6 | 71.5 | 90.4 | 93.7 | 94.2 |
| Zipper | 97.5 | 97.1 | 99.7 | 99.6 | 99.3 |
| Tile | 100.0 | 100.0 | 99.9 | 100.0 | 99.8 |
| Wood | 99.8 | 99.7 | 99.4 | 99.2 | 99.6 |
| Mean | 96.5 | 93.5 | 94.8 | 96.2 | 96.7 |

Table 13: Subset-level performance comparison (AUROC) for anomaly segmentation on VisA.

| Object name | WinCLIP | VAND | CoOp | AnomalyCLIP | TokenCLIP |
|---|---|---|---|---|---|
| Candle | 88.9 | 97.8 | 16.3 | 98.8 | 98.8 |
| Capsules | 81.6 | 97.5 | 47.5 | 95.0 | 95.4 |
| Cashew | 84.7 | 86.0 | 32.5 | 93.8 | 94.5 |
| Chewinggum | 93.3 | 99.5 | 3.4 | 99.3 | 99.6 |
| Fryum | 88.5 | 92.0 | 21.7 | 94.6 | 94.8 |
| Macaroni1 | 70.9 | 98.8 | 36.8 | 98.3 | 98.8 |
| Macaroni2 | 59.3 | 97.8 | 27.5 | 97.6 | 98.4 |
| Pcb1 | 61.2 | 92.7 | 19.8 | 94.1 | 95.4 |
| Pcb2 | 71.6 | 89.7 | 22.9 | 92.4 | 92.4 |
| Pcb3 | 85.3 | 88.4 | 18.0 | 88.4 | 87.9 |
| Pcb4 | 94.4 | 94.6 | 14.0 | 95.7 | 95.6 |
| Pipe_fryum | 75.4 | 96.0 | 29.2 | 98.2 | 99.1 |
| Mean | 79.6 | 94.2 | 24.2 | 95.5 | 95.9 |

Table 14: Subset-level performance comparison (PRO) for anomaly segmentation on VisA.

| Object name | WinCLIP | VAND | CoOp | AnomalyCLIP | TokenCLIP |
|---|---|---|---|---|---|
| Candle | 83.5 | 92.5 | 1.1 | 96.2 | 95.5 |
| Capsules | 35.3 | 86.7 | 18.4 | 78.5 | 80.3 |
| Cashew | 76.4 | 91.7 | 1.7 | 91.6 | 95.1 |
| Chewinggum | 70.4 | 87.3 | 0.1 | 91.2 | 92.8 |
| Fryum | 77.4 | 89.7 | 2.6 | 86.8 | 88.3 |
| Macaroni1 | 34.3 | 93.2 | 18.1 | 89.8 | 92.4 |
| Macaroni2 | 21.4 | 82.3 | 2.7 | 84.2 | 88.4 |
| Pcb1 | 26.3 | 87.5 | 0.1 | 81.7 | 86.8 |
| Pcb2 | 37.2 | 75.6 | 0.7 | 78.9 | 80.5 |
| Pcb3 | 56.1 | 77.8 | 0.0 | 77.1 | 75.9 |
| Pcb4 | 80.4 | 86.8 | 0.0 | 91.3 | 90.1 |
| Pipe_fryum | 82.3 | 90.9 | 0.6 | 96.8 | 96.7 |
| Mean | 56.8 | 86.8 | 3.8 | 87.0 | 88.5 |

Table 15: Subset-level performance comparison (AUROC) for anomaly classification on VisA.

| Object name | WinCLIP | VAND | CoOp | AnomalyCLIP | TokenCLIP |
|---|---|---|---|---|---|
| Candle | 95.4 | 83.8 | 46.2 | 79.3 | 85.6 |
| Capsules | 85.0 | 61.2 | 77.2 | 81.5 | 90.9 |
| Cashew | 92.1 | 87.3 | 75.7 | 76.3 | 92.4 |
| Chewinggum | 96.5 | 96.4 | 84.9 | 97.4 | 98.2 |
| Fryum | 80.3 | 94.3 | 80.0 | 93.0 | 95.7 |
| Macaroni1 | 76.2 | 71.6 | 53.6 | 87.2 | 85.0 |
| Macaroni2 | 63.7 | 64.6 | 66.5 | 73.4 | 78.2 |
| Pcb1 | 73.6 | 53.4 | 24.7 | 85.4 | 71.1 |
| Pcb2 | 51.2 | 71.8 | 44.6 | 62.2 | 67.1 |
| Pcb3 | 73.4 | 66.8 | 54.4 | 62.7 | 73.1 |
| Pcb4 | 79.6 | 95.0 | 66.0 | 93.9 | 96.5 |
| Pipe_fryum | 69.7 | 89.9 | 80.1 | 92.4 | 98.0 |
| Mean | 78.1 | 78.0 | 62.8 | 82.1 | 85.8 |

Table 16: Subset-level performance comparison (AP) for anomaly classification on VisA.

| Object name | WinCLIP | VAND | CoOp | AnomalyCLIP | TokenCLIP |
|---|---|---|---|---|---|
| Candle | 95.8 | 86.9 | 52.9 | 81.1 | 86.9 |
| Capsules | 90.9 | 74.3 | 85.3 | 88.7 | 95.7 |
| Cashew | 96.4 | 94.1 | 87.1 | 89.4 | 96.8 |
| Chewinggum | 98.6 | 98.4 | 93.1 | 98.9 | 99.4 |
| Fryum | 90.1 | 97.2 | 90.2 | 96.8 | 98.3 |
| Macaroni1 | 75.8 | 70.9 | 52.3 | 86.0 | 86.8 |
| Macaroni2 | 60.3 | 63.2 | 62.2 | 72.1 | 78.4 |
| Pcb1 | 78.4 | 57.2 | 36.0 | 87.0 | 74.4 |
| Pcb2 | 49.2 | 73.8 | 47.3 | 64.3 | 67.6 |
| Pcb3 | 76.5 | 70.7 | 54.8 | 70.0 | 78.5 |
| Pcb4 | 77.7 | 95.1 | 66.3 | 94.4 | 96.4 |
| Pipe_fryum | 82.3 | 94.8 | 89.7 | 96.3 | 99.0 |
| Mean | 81.2 | 81.4 | 68.1 | 85.4 | 88.2 |

# J  ADDITIONAL THEORETICAL DETAILS

The two components of the subspace mixture inequality equation 10 in the main text: (i) the *subspace specialization cost* arising when a subspace aligns to a single visual cluster, and (ii) the *mixed-cluster penalty* incurred when a subspace receives mass from multiple clusters.

## A.1  COSINE COST AND THE OT OBJECTIVE

In CLIP, both visual features $v_i$ and subspace vectors $o_j$ are $\ell_2$-normalized. The cost in balanced OT is the cosine distance

$$C_{ij} = 1 - \langle v_i, o_j \rangle.$$

Using $\|v_i\| = \|o_j\| = 1$, we expand

$$\|v_i - o_j\|^2 = \|v_i\|^2 + \|o_j\|^2 - 2\langle v_i, o_j \rangle = 2(1 - \langle v_i, o_j \rangle) = 2C_{ij},$$

which gives the equivalence

$$C_{ij} = \tfrac{1}{2}\|v_i - o_j\|^2.$$

Thus, up to an irrelevant constant factor, the balanced OT objective reduces to minimizing a squared Euclidean alignment loss:

$$\mathcal{L}_{\mathrm{OT}} = \sum_{i=1}^{N}\sum_{j=1}^{Q} T_{ij}\, \|v_i - o_j\|^2, \qquad T \in \Pi(u, v).$$

## A.2  BIAS–VARIANCE DECOMPOSITION AND SPECIALIZATION COST

Let $\mathcal{C}_k$ be a visual cluster with centroid $\mu_k$ and variance

$$\sigma_k^2 = \frac{1}{|\mathcal{C}_k|}\sum_{i \in \mathcal{C}_k} \|v_i - \mu_k\|^2.$$

**Lemma J.1.** *For any $o \in \mathbb{R}^d$,*

$$\frac{1}{|\mathcal{C}_k|}\sum_{i \in \mathcal{C}_k} \|v_i - o\|^2 = \sigma_k^2 + \|\mu_k - o\|^2.$$

This identity shows that the alignment cost decomposes into 1) an irreducible intra-cluster term $\sigma_k^2$, and 2) a geometric deviation term $\|\mu_k - o\|^2$. Hence, if a subspace $o_j$ *specializes* to cluster $\mathcal{C}_k$, the optimal choice is $o_j = \mu_k$, and the minimal achievable alignment cost is

$$\min_{o_j}\sum_{i \in \mathcal{C}_k} T_{ij}\|v_i - o_j\|^2 = \alpha_k\, \sigma_k^2, \qquad \alpha_k = \sum_{i \in \mathcal{C}_k} T_{ij}.$$

Because balanced OT enforces $v_j > 0$, every subspace must absorb a positive amount of mass.

## A.3  SUBSPACE MIXTURE PENALTY

We now derive the additional cost incurred when a subspace mixes two clusters. Let $\mathcal{C}_p$ and $\mathcal{C}_q$ have transported masses

$$\alpha_p = \sum_{i \in \mathcal{C}_p} T_{ij} > 0, \qquad \beta_q = \sum_{i \in \mathcal{C}_q} T_{ij} > 0,$$

and treat $T_{ij}/\alpha_p$ and $T_{ij}/\beta_q$ as probability weights. Applying Lemma J.1 in this weighted form gives

$$\sum_{i \in \mathcal{C}_p} T_{ij}\|v_i - o_j\|^2 = \alpha_p\sigma_p^2 + \alpha_p\|\mu_p - o_j\|^2,$$

$$\sum_{i \in \mathcal{C}_q} T_{ij}\|v_i - o_j\|^2 = \beta_q\sigma_q^2 + \beta_q\|\mu_q - o_j\|^2.$$

Summing yields the exact decomposition

$$\underbrace{\sum_{i \in \mathcal{C}_p} T_{ij} \|v_i - o_j\|^2 + \sum_{i \in \mathcal{C}_q} T_{ij} \|v_i - o_j\|^2}_{\text{Subspace mixture cost}} = \underbrace{\alpha_p \sigma_p^2 + \beta_q \sigma_q^2}_{\text{Subspace specialization cost}} + \underbrace{\left( \alpha_p \|\mu_p - o_j\|^2 + \beta_q \|\mu_q - o_j\|^2 \right)}_{\text{Penalty}}.$$

To bound the offset, we apply the mixed-cluster penalty:

**Lemma J.2.** *Let* $D_{pq} = \|\mu_p - \mu_q\| > 0$. *Then for any* $o_j$,

$$\alpha_p \|\mu_p - o_j\|^2 + \beta_q \|\mu_q - o_j\|^2 \geq \frac{\alpha_p \beta_q}{\alpha_p + \beta_q} D_{pq}^2.$$

Substituting this bound gives the subspace mixture inequality

$$\sum_{i \in \mathcal{C}_p} T_{ij} \|v_i - o_j\|^2 + \sum_{i \in \mathcal{C}_q} T_{ij} \|v_i - o_j\|^2 \geq \alpha_p \sigma_p^2 + \beta_q \sigma_q^2 + \frac{\alpha_p \beta_q}{\alpha_p + \beta_q} \|\mu_p - \mu_q\|^2.$$

This establishes Equation 10 in the main text.