# OpenReview forum: "TokenCLIP: Token-wise Prompt Learning for Zero-shot Anomaly Detection"
_ICLR.cc/2026/Conference — Submitted to ICLR 2026_

### Official Review · Reviewer_foti · 2025-10-28

**Soundness:** 3
**Presentation:** 3
**Contribution:** 3
**Rating:** 6
**Confidence:** 4

**Summary:**

This paper proposes a novel zero-shot anomaly detection method, TokenCLIP, which consists of four key components: base semantics learning, decoupled local and global text prompts, and orthogonal regularization. The proposed approach demonstrates superior performance on both industrial and medical datasets, achieving high detection accuracy while maintaining fast inference speed.

**Strengths:**

1.The motivation of the paper is clear. To address the problem of indiscriminate alignment in existing methods, the authors introduce multiple orthogonal textual subspaces to dynamically align each visual patch token according to its visual semantics.
2.The experiments are comprehensive. The proposed method is validated on both industrial and medical datasets, achieving state-of-the-art results in anomaly detection and localization. Furthermore, the comparison of inference efficiency convincingly shows the method’s high effectiveness.
3.The ablation study is well-designed and thoroughly demonstrates the contribution of each component.
4.The overall structure of the paper is clear and logically organized.

**Weaknesses:**

1.The paper proposes multiple orthogonal textual subspaces to address the issue of indiscriminate alignment, which is an interesting and promising idea. However, the specific implementation details and the underlying rationale for this design are not clearly explained in the current version of the paper.
2.The paper lacks sufficient discussion on the selection and sensitivity of key hyperparameters involved in this process.
3.There are several minor typographical issues in the manuscript. For instance, in the first paragraph of the Introduction, the sentence explore zero-shot capabilities by adapting FMs ((Pang et al., 2021; Zhou et al., 2022; Khattak et al., 2023; Jeong et al., 2023; Zhou et al., 2024a; ?) contains an extra question mark. Similarly, another question mark appears in These methods typically project either learnable (Zhou et al., 2024a; ?) in the second paragraph. These should be corrected for clarity.

**Questions:**

1.It is not entirely clear how the orthogonality of the textual subspaces obtained via multi-head projection is verified. It would be helpful to provide a more intuitive and concrete illustration—such as a visualization (e.g., t-SNE plot)—to support this claim.
2.The paper states that TokenCLIP benefits most from an appropriate k that supports subspace specialization while avoiding semantic over-coupling. As shown in Table 6, model performance first improves and then declines as k increases. Given this, how should the hyperparameter k and the subspace number Q be selected in practical applications?
3.From Table 4, the improvement brought by semantics learning is much more significant on the MVTec dataset than on the Visa dataset. Could the authors elaborate on the reasons behind this large performance discrepancy?

---

> ### Author Response · Authors · 2025-11-23
>
> **Q1. The specific implementation details and the underlying rationale for this design are not clearly explained in the current version of the paper.**
>
> Thank you for the insightful comments. As shown in the ablation study, TokenCLIP is robust to hyperparameter choices, and we do not manually tune them for performance gains. Instead, all parameters are selected via a straightforward grid search. Regarding the rationale behind using a fine-grained textual space, we provide a theoretical explanation in the **general response** on how minimizing the OT objective induces semantic specialization. In the revision, we further include a formal justification in Theorem 3.1, which establishes that the OT objective penalizes subspace mixing and therefore leads to specialization.
>
> **Q2.The paper lacks sufficient discussion on the selection and sensitivity of key hyperparameters involved in this process.**
>
> Thank you for pointing out this question. In the original version, we already provided and discussed key hyperparameters, including the subspace number (Table 5), top-K (Table 6), threshold (Table 8), and the coefficients of the hinge loss and regularization terms (Figures 5 and 6). These hyperparameters are primarily introduced by the optimal transport formulation. In practice, we select them via simple grid search, without tuning them specifically for performance. We also observe that TokenCLIP is highly robust to these choices, and its performance remains stable across a wide range of values.
>
> Intuitively, the subspace number controls the granularity of decomposition, while top-K and threshold determine the purity of each subspace. Our method is robust to different combinations of these settings. For instance, we use 4 subspaces on MVTec and 3 subspaces on VisA; swapping these choices (subspace = 3 for MVTec or subspace = 4 for VisA) yields only minor performance changes (Table 5), demonstrating that the model is not sensitive to this parameter.
>
> **Q3. It is not entirely clear how the orthogonality of the textual subspaces obtained via multi-head projection is verified. It would be helpful to provide a more intuitive and concrete illustration—such as a visualization (e.g., t-SNE plot).**
>
> Thanks for your comments. We have supplemented the visualization of subspace textual orthogonality in Appendix F. We clarify that the purpose of the orthogonality regularization is not to create specialization, but to prevent the specialized textual subspaces from becoming excessively overlapping. The textual specialization in TokenCLIP is primarily induced by OT. Importantly, achieving strong specialization does not require the subspaces to be strictly orthogonal. Each subspace is intended to align with a distinct visual semantic pattern, and since these visual semantics are inherently correlated, the corresponding textual subspaces only need sufficient separation, not perfect orthogonality.
>
> To verify this, we compute the cosine similarities between the learned textual subspaces for both the OT-enabled model and the variant without OT. Note that values close to 1 indicate near-parallel vectors, while values close to 0 indicate near-orthogonal vectors. As shown in Figure 10 (in the revision), abnormality textual subspaces in TokenCLIP with OT are less orthogonal than those without OT. However, the latter fails to learn meaningful specialized semantics, as also demonstrated in Figure 5 (in the revision). This confirms that orthogonality is not the main driver of specialization. Rather, OT is the key factor, and the regularization serves to further improve specialization.
>
> **Q4. The paper states that TokenCLIP benefits most from an appropriate k that supports subspace specialization while avoiding semantic over-coupling. As shown in Table 6, model performance first improves and then declines as k increases. Given this, how should the hyperparameter k and the subspace number Q be selected in practical applications?**
>
> Thanks for your insightful comments. These hyperparameters determine the semantic diversity that can be captured from the auxiliary dataset. If the auxiliary dataset contains a larger variety of categories and object semantics, using a larger number of textual subspaces (**Q**) allows the model to better partition and represent these semantics. This is the reason why we use 4 subspaces on MVTec (more object categories) and 3 subspaces on VisA.
>
> Regarding the choice of **k**, setting **k** = 2 is generally appropriate: it provides sufficient capacity for subspace specialization while avoiding over-coupling between the learned semantics. We found this setting to be stable and effective across all datasets.

---

> > ### Author Response · Authors · 2025-11-23
> >
> > **Q5. From Table 4, the improvement brought by semantics learning is much more significant on the MVTec dataset than on the VisA dataset. Could the authors elaborate on the reasons behind this large performance discrepancy?**
> >
> > Thanks for your insightful comments. Previous works rely on a single textual subspace to capture anomaly semantics in the target dataset. When the target dataset contains rich and diverse semantics, this single-subspace design exhibits more pronounced limitations.
> >
> > In contrast, TokenCLIP learns fine-grained semantics through multiple specialized subspaces. Therefore, the performance gains are naturally larger when the target dataset contains greater semantic diversity. Since MVTec includes richer semantics than VisA (i.e., more object categories), the improvement achieved by TokenCLIP is correspondingly more pronounced on MVTec.

---

### Official Review · Reviewer_i1YR · 2025-10-30

**Soundness:** 3
**Presentation:** 3
**Contribution:** 3
**Rating:** 6
**Confidence:** 4

**Summary:**

This paper presents TokenCLIP, a novel framework for ZSAD that addresses the limitations of existing CLIP-based methods. The authors argue that the standard approach of aligning all visual tokens with a single, indiscriminate textual embedding compromises the model's ability to capture diverse anomaly semantics. To overcome this, TokenCLIP introduces a dynamic, token-level alignment mechanism. The core idea is to project a base textual space into multiple orthogonal subspaces and then use an Optimal Transport formulation to assign each visual token to a weighted combination of these subspaces. This approach allows for more fine-grained supervision, enabling the model to learn specialized representations for different semantic patterns. The paper demonstrates through extensive experiments on industrial and medical datasets that TokenCLIP achieves state-of-the-art performance.

**Strengths:**

1. The paper identifies a clear and significant limitation in current ZSAD methods and proposes a well-motivated and elegant solution.
2. The application of Optimal Transport to dynamically align visual tokens with a set of learned subspaces is a novel contribution to the field of anomaly detection. This formulation provides a principled way to achieve fine-grained, many-to-many correspondence.
3. The experimental results are strong and comprehensive. The method shows consistent and significant performance gains over strong baselines across a wide variety of 15 different datasets, demonstrating its effectiveness and generalizability.

**Weaknesses:**

1. The mechanism that drives the semantic specialization of the subspaces could be explained more clearly. The paper attributes this to the minimal cost objective of OT. However, OT's primary role is to find the most efficient matching between two fixed distributions. The specialization itself may heavily rely on the orthogonality regularization term $L_{reg}$, which explicitly forces the subspaces to be distinct. The paper would be stronger if it disentangled the contribution of OT's cost minimization from that of the explicit regularization in achieving this specialization.
2. While the authors provide an analysis of computational overhead, the discussion is brief. The introduction of an iterative OT solver for every batch during training can be computationally demanding. A more detailed analysis comparing the training time and convergence speed against baselines like AnomalyCLIP would be beneficial for practitioners assessing the method's trade-offs.

**Questions:**

1. The motivation for the method hinges on using multiple "textual subspaces." Have the authors considered whether these subspaces are fundamentally different from a set of randomly initialized, learnable vectors? In other words, if the "base textual space" was replaced with a generic learnable parameter, could the model still achieve comparable performance through the OT and regularization framework? This would help clarify the importance of starting from a text-based embedding.
2. How does the OT formulation specifically encourage different subspaces to focus on different semantics (e.g., one on foreground objects, others on background textures)? The paper suggests this is a result of the minimal cost objective. Could the authors elaborate on the intuition behind this? Is it possible that without the orthogonality constraint L_reg, a single "winner-take-all" subspace could still dominate the alignment for most tokens, thus preventing specialization?

---

> ### Author Response · Authors · 2025-11-23
>
> We thank you for your efforts in reviewing our paper.
>
> **Q1. The paper would be stronger if it disentangled the contribution of OT's cost minimization from that of the explicit regularization in achieving this specialization.**
>
> Thank you for the insightful comment. We have expanded subsection 4.2 (“What makes the textual specialization”) to provide a more detailed analysis of how OT and the regularization term contribute to subspace specialization. As shown in Figure 5 (in the revision), OT plays the dominant role in enabling textual subspace specialization: even without the regularization term, TokenCLIP can still achieve specialization when OT is present. In contrast, when OT is removed, the regularization term alone fails to produce meaningful specialization. Nevertheless, the regularization term can assist OT in further enhancing the degree of specialization.
>
> **Q2. The introduction of an iterative OT solver for every batch during training can be computationally demanding. A more detailed analysis comparing the training time and convergence speed against baselines like AnomalyCLIP would be beneficial for practitioners assessing the method's trade-offs.**
>
> Thanks for the insightful comments. We have added the new metric “training time per epoch” in Table 3 of the revision. The iterative OT step is lightweight: each iteration only requires two element-wise division operations to update the vectors u and v, so the computation cost is minimal and does not slow down training. As a result, the overall training speed remains efficient. The training times of AnomalyCLIP, FAPrompt, and TokenCLIP are 96s, 245s, and 110s, respectively. Although TokenCLIP incurs slightly more overhead, it achieves significantly better performance.
>
> Compared to the ensemble-prompt design in FAPrompt, TokenCLIP uses a multi-head projection to generate multiple textual subspaces from a shared base textual embedding, instead of encoding each subspace separately with the text encoder. This avoids repeatedly forwarding the text encoder, which would otherwise introduce substantial computational cost.
>
> **Q3. Have the authors considered whether these subspaces are fundamentally different from a set of randomly initialized, learnable vectors?**
>
> Thanks for your comments. We have added the analysis on the difference between subspaces with randomly initialized learnable vectors in Appendix G of the revision. To investigate this, we remove the text encoder and initialize the subspaces as learnable random vectors while keeping the rest of the framework unchanged. We denote this variant as TokenCLIP-Visual.
>
> On MVTec AD, TokenCLIP-Visual achieves 84.5 AUROC and 82.3 PRO, which are significantly lower than the full TokenCLIP (92.2 AUROC and 87.9 PRO). This degradation stems from single-modal alignment: removing the text encoder causes the learnable subspaces to function merely as clusters over the visual semantics extracted from the auxiliary data. In this case, alignment performs purely in a visual–visual space. However, CLIP’s strong zero-shot generalization derives from its joint visual–text embedding space, learned via large-scale contrastive pretraining. TokenCLIP-Visual loses this crucial cross-modal alignment, making it less capable of generalizing to unseen categories under zero-shot settings.
>
> In contrast, TokenCLIP fine-grains the textual space while preserving CLIP’s cross-modal structure, thereby inheriting its strong zero-shot generalization capability.
>
> **Q4. How does the OT formulation specifically encourage different subspaces to focus on different semantics (e.g., one on foreground objects, others on background textures)? The paper suggests this is a result of the minimal cost objective.**
>
> Thanks for your insightful comments. We provide a theoretical analysis of why minimizing the OT cost naturally leads to textual specialization in the general response above. In the revision, we additionally include a formal explanation in Theorem 3.1, which shows that minimizing the OT objective penalizes subspace mixing and consequently induces specialization. This provides a principled justification for why OT is effective in guiding the textual subspaces to become specialized.
>
> **Q5. Is it possible that without the orthogonality constraint L_reg, a single "winner-take-all" subspace could still dominate the alignment for most tokens, thus preventing specialization?**
>
> Thanks for your comments. Please refer to Q1.

---

### Official Review · Reviewer_5gCu · 2025-10-31

**Soundness:** 2
**Presentation:** 2
**Contribution:** 3
**Rating:** 4
**Confidence:** 5

**Summary:**

This paper addresses the task of ZSAD using CLIP-based models. The authors identify a limitation in current methods, which typically align all visual tokens from an image to a single, shared textual embedding space. They propose a method, TokenCLIP, which instead performs a token-wise alignment. The proposed framework expands a base textual space into a set of orthogonal subspaces using a multi-head projection. The core of the method is the formulation of a dynamic alignment mechanism as an Optimal Transport problem, which maps each visual token to a combination of these textual subspaces based on a cross-modal similarity cost. The resulting transport plan is sparsified using a top-K masking approach to produce the final assignment weights. The model is trained end-to-end with a combination of losses, including a global image-level loss, a base local loss from an initial indiscriminate alignment, and a dynamic alignment loss derived from the OT plan. The effectiveness of TokenCLIP is evaluated on 15 industrial and medical datasets for both image-level anomaly detection and pixel-level anomaly segmentation.

**Strengths:**

1.The proposed method demonstrates SOTA performance across multiple datasets.

2.The central idea of employing textual subspaces and formulating the alignment as an Optimal Transport problem is novel for this task.

**Weaknesses:**

1.The claim that the learned spaces are "textual subspaces" is not sufficiently justified with theoretical or experimental evidence. CLIP-based zero-shot anomaly detection operates on the premise that CLIP has aligned visual features with semantic textual features during pre-training. Detection is achieved by comparing image features against textual embeddings that explicitly represent concepts like "normal" and "abnormal." However, in this work, the embeddings within the so-called "textual subspaces" are derived from initial text prompts but are then optimized via a multi-head projection and end-to-end training. It becomes unclear whether these final embeddings still retain their original textual semantics. Are they still representative of textual information, or have they become latent vectors that are simply trained to have high similarity with normal or abnormal visual features? This ambiguity raises questions about the fundamental working principle of the method. It is possible that the model is learning a direct mapping from visual features to classifiable embeddings, rather than leveraging the rich, generalizable text-image alignment that is the cornerstone of CLIP's zero-shot capabilities.

2.While the paper visualizes the spatial assignments of subspaces, the semantic roles of these subspaces are not deeply analyzed. The interpretation that one subspace captures "object-centric semantics" while others capture "background" is based on visual inspection. A more quantitative analysis linking subspaces to specific anomaly categories, object parts, or textual concepts would provide stronger evidence for semantic specialization.

3.There appears to be missing citations in the introduction (line 037). The citation list reads "...Zhou et al., 2024a; ?", which should be corrected.

**Questions:**

1.In Table 5, the case where Q=1 seems conceptually similar to the baseline AnomalyCLIP, as both use a single textual space for alignment. However, the results for Q=1 show a significant performance improvement over the reported AnomalyCLIP baseline. Could the authors clarify the architectural or training differences that lead to this performance gain? Is the Q=1 model not exactly equivalent to AnomalyCLIP?

2.According to Figure 3(c), on the MVTec AD dataset, the TokenCLIP-Van variant (which uses greedy local matching instead of OT) results in a 4.2 point drop in Image-level AUROC compared to the full TokenCLIP model. This performance drop would place TokenCLIP-Van significantly below the AnomalyCLIP baseline, which uses only a single textual space. This result is counterintuitive, as one might expect that ensembling multiple textual features, even with a simple greedy assignment, would not perform worse than using a single feature space. What is the authors' explanation for this phenomenon? Why does the introduction of multiple subspaces without the global optimization of OT lead to a performance degradation compared to the simpler single-subspace baseline?

---

> ### Author Response · Authors · 2025-11-23
>
> We thank you for your efforts in reviewing our paper.
>
> **Q1. Are they still representative of textual information, or have they become latent vectors that are simply trained to have high similarity with normal or abnormal visual features?**
>
> Thanks for your insightful comments. We fine-grain the original textual space into
> multiple subspaces, each specializing in different semantics. The base text
> prompts already encode generalized semantics, so we only need to operate on
> this foundational textual space. This is why we name them “textual subspaces”
> instead of textual prompts.
>
> We provide demonstrations from two aspects as follows:
>
> - We have added the analysis on the difference between subspaces with randomly initialized learnable vectors in Appendix G of the revision. To investigate this, we remove the text encoder and initialize the subspaces as learnable random vectors while keeping the rest of the framework unchanged. We denote this variant as TokenCLIP-Visual. On MVTec AD, TokenCLIP-Visual achieves 84.5 AUROC and 82.3 PRO, which are significantly lower than the full TokenCLIP (92.2 AUROC and 87.9 PRO). This degradation stems from single-modal alignment: removing the text encoder causes the learnable subspaces to function merely as clusters over the visual semantics extracted from the auxiliary data. In this case, alignment performs purely in a visual–visual space. However, CLIP’s strong zero-shot generalization derives from its joint visual–text embedding space, learned via large-scale contrastive pretraining. TokenCLIP-Visual loses this crucial cross-modal alignment, making it less capable of generalizing to unseen categories under zero-shot settings. In contrast, TokenCLIP fine-grains the textual space while preserving CLIP’s cross-modal structure, thereby inheriting its strong zero-shot generalization capability.
>
> - In the revision, we have expanded subsection 4.2 “What makes the textual specialization” with additional analysis and visualization. Figure 5 now illustrates the subspace assignment frequency over foreground (FG) and background (BG) regions across categories. The results show that one subspace consistently attends to object foregrounds, while the rest subspaces focus on background regions. This separation confirms that TokenCLIP effectively learns specialized and interpretable subspaces.
>
> **Q2. A more quantitative analysis linking subspaces to specific anomaly categories, object parts, or textual concepts would provide stronger evidence for semantic specialization.**
>
> Thank you for the insightful comments. In the revision, we have expanded subsection 4.2 “What makes the textual specialization” with additional analysis and visualization. Figure 5 now illustrates the subspace assignment frequency over foreground (FG) and background (BG) regions across categories. The results show that one subspace consistently attends to object foregrounds, while the rest subspaces focus on background regions. This separation confirms that TokenCLIP effectively learns specialized and interpretable subspaces.
>
> **Q3.There appears to be missing citations in the introduction (line 037). The citation list reads "...Zhou et al., 2024a; ?", which should be corrected.**
>
> Thank you for your helpful feedback. We have corrected the rendering question about the citation issues in the introduction. And we have thoroughly proofread the entire manuscript.
>
> **Q4. However, the results for **Q**=1 show a significant performance improvement over the reported AnomalyCLIP baseline. Could the authors clarify the architectural or training differences that lead to this performance gain? Is the **Q**=1 model not exactly equivalent to AnomalyCLIP?**
>
> Thank you for your comments. We would like to clarify that when **Q**=1, TokenCLIP is not equivalent to AnomalyCLIP. Beyond the dynamic alignment mechanism, TokenCLIP explicitly decouples the learning of global and local anomaly semantics (Section 3.1), which alleviates the semantic coupling issue inherent in AnomalyCLIP. Moreover, TokenCLIP incorporates a hinge loss that promotes a more precise and discriminative decision boundary. These design differences collectively explain why TokenCLIP outperforms AnomalyCLIP even under the **Q**=1 setting.

---

> > ### Author Response · Authors · 2025-11-23
> >
> > **Q5. Why does the introduction of multiple subspaces without the global optimization of OT lead to a performance degradation compared to the simpler single-subspace baseline?**
> >
> > Thanks for your comments. We provide a detailed explanation in Section 4.2 (“OT is important for dynamic alignment”). Without the constraints imposed by OT, the assignment of visual tokens to textual subspaces degenerates into a greedy allocation: one dominant subspace absorbs most visual tokens during training, while the remaining subspaces receive very little tokens and thus suffer from severely under-optimized. These under-optimized subspaces fail to capture meaningful normal/abnormal semantics.
> >
> > In zero-shot anomaly detection, where unseen objects appear at test time,
> > distribution shifts between the auxiliary training data and the test categories
> > may cause the model to select poorly optimized subspaces. This degrades
> > performance and can even result in worse accuracy than using a single subspace.
> >
> > In contrast, the single-subspace model does not suffer from this issue because it consistently relies on the same subspace for both training and inference. This explains why removing OT may lead to worse performance compared to using a single subspace.

---

### Official Review · Reviewer_coh7 · 2025-11-02

**Soundness:** 3
**Presentation:** 3
**Contribution:** 2
**Rating:** 6
**Confidence:** 3

**Summary:**

The paper proposes TokenCLIP, a fine-grained alignment framework that adaptively assigns a weighted combination of textual subspaces to each visual token. It reformulates dynamic alignment between tokens and orthogonal textual subspaces as an optimal transport (OT) problem, which helps ensure sufficient optimization and encourages semantic specialization across subspaces. A top-K masking is applied to further sparsify the transport plan and specialize subspaces for distinct visual regions. Extensive experiments validate the method across multiple datasets.

**Strengths:**

1. The method is well designed, with a clear and coherent architecture

2. The writing is fluent and clear.

3. The experimental coverage is fairly comprehensive, spanning multiple datasets.

**Weaknesses:**

1. No localization visualizations are provided, only numerical results, which weakens the credibility of the experiments.

2. There are many hyperparameters; does performance require per-dataset tuning?

3. Some implementation details are missing, e.g., OT marginals and weight settings are insufficiently specified.

4. There are several minor errors, such as multiple citations in the introduction rendered as “?”.

**Questions:**

See weaknesses:

1. No localization visualizations are provided, only numerical results, which weakens the credibility of the experiments.

2. There are many hyperparameters; does performance require per-dataset tuning?

3. Some implementation details are missing, e.g., OT marginals and weight settings are insufficiently specified.

4. There are several minor errors, such as multiple citations in the introduction rendered as “?”.

---

> ### Author Response · Authors · 2025-11-23
>
> We thank you for your efforts in reviewing our paper.
>
> **Q1. No localization visualizations are provided, only numerical results, which weakens the credibility of the experiments.**
>
> Thank you for your detailed feedback. We have included visualizations of both
> TokenCLIP and its counterpart TokenCLIP-Van without OT in Appendix E in revision.
>
> **Q2. There are many hyperparameters; does performance require per-dataset tuning?**
>
> Thank you for your insightful feedback. These parameters are mainly introduced by the optimal transport formulation. In practice, we select them using a standard grid search. The model is robust to these hyperparameters, and its performance remains stable across a wide range of values.
>
> For example, we set the number of subspaces to 4 on MVTec and 3 on VisA. As shown in Table 5, swapping these choices (i.e., using 3 subspaces for MVTec or 4 subspaces for VisA) leads to only minor performance changes. This demonstrates that TokenCLIP is not sensitive to the choice of this parameter.
>
> **Q3. Some implementation details are missing, e.g., OT marginals and weight settings are insufficiently specified.**
>
> Thank you for your detailed feedback. We clarify here the implementation details of the OT marginals.
>
> The marginal vectors **u** and **v** have dimensions N (number of visual tokens) and Q (number of textual subspaces), respectively. Following common practice, both marginals are initialized as uniform distributions. Accordingly, their weights are set to:
> $$ p_i = \frac{1}{N} $$
> and
> $$ q_j = \frac{1}{Q} $$
>
> We include these details in the implementation details in revision.
>
> **Q4. There are several minor errors, such as multiple citations in the introduction rendered as “?”.**
>
> Thank you for your helpful feedback. We have corrected the rendering question about the citation issues in the introduction. And we have thoroughly proofread the entire manuscript.

---

### Author Response · Authors · 2025-11-23
**General Response**

# General Response

Dear Reviewers and ACs,

We sincerely appreciate the detailed reviews and the constructive, positive feedback provided by all reviewers. While we address each comment individually, we also provide here a theoretical explanation of why the optimal transport (OT) objective induces subspace specialization.

**Theorem: OT penalizes subspace mixture and induces specialization**

For $\ell_2$-normalized visual features $v_i$ and textual subspaces $o_j$, the OT cost between a visual token and a subspace is defined using the cosine distance. Since both vectors are $\ell_2$-normalized, we have:

$$
1 - \langle v_i, o_j \rangle = \tfrac{1}{2}\|v_i - o_j\|^2
$$

Therefore, minimizing the cosine-distance OT objective is equivalent to minimizing a squared Euclidean-distance OT objective. The balanced OT loss becomes:

$$ \mathcal{L}_ {\mathrm{OT}}= \sum_{i=1}^N \sum_{j=1}^Q T_{ij}\|v_i - o_j\|^2, \quad T \in \Pi(u,v). $$

Here, $T_{ij}$ is the transport plan between visual token $v_i$ and subspace $o_j$, and $\Pi(u,v)$ is the set of couplings with fixed marginals.

---

To understand why OT encourages subspace specialization, consider two visual clusters $\mathcal{C}_p$ and $\mathcal{C}_q$ with centroids $\mu_p$ and $\mu_q$, and variances:

$$\sigma_ p^2 = \frac{1}{|\mathcal{C}_ p|} \sum_ {i\in\mathcal{C}_ p} \|v_i - \mu_ p\|^2$$

$$\sigma_ q^2 = \frac{1}{|\mathcal{C}_ q|} \sum_{i\in\mathcal{C}_ q} \|v_i - \mu_q\|^2$$

Suppose a subspace $o_j$ receives mass from both clusters, with transported amounts $\alpha_p > 0$ from $\mathcal{C}_p$ and $\beta_q > 0$ from $\mathcal{C}_q$. Its OT cost is:

$$
\sum_ {i\in\mathcal{C}_ p} T_{ij}\|v_i - o_j\|^2
+
\sum_ {i\in\mathcal{C}_ q} T_{ij}\|v_i - o_j\|^2
$$

Using variance decomposition, this mixture cost can be lower-bounded by:

---

**1. A within-cluster specialization cost**

$$
\alpha_p \sigma_p^2 + \beta_q \sigma_q^2
$$

**2. A between-cluster penalty**

$$
\frac{\alpha_p\beta_q}{\alpha_p+\beta_q}\|\mu_p - \mu_q\|^2
$$

---

Putting it together:

$$
\underbrace{
\sum_ {i\in\mathcal{C}_ p} T_ {ij}\|v_i - o_j\|^2
+
\sum_ {i\in\mathcal{C}_ q} T_ {ij}\|v_i - o_j\|^2
}_ {\text{Subspace mixture cost}}
\ge
\underbrace{
\alpha_p \sigma_p^2 + \beta_q \sigma_q^2
}_ {\text{Subspace specialization cost}}
+
\underbrace{
\frac{\alpha_p\beta_q}{\alpha_p+\beta_q}\|\mu_p - \mu_q\|^2
}_{\text{Penalty}}
$$

Equality holds **if and only if** $
\mu_p = \mu_q ,
$ i.e., **two clusters have identical means and therefore encode the same semantics**.

When $\mu_p \neq \mu_q$ (from two clusters), the penalty term $
\frac{\alpha_p \beta_q}{\alpha_p + \beta_q}\|\mu_p - \mu_q\|^2 > 0
$ is strictly positive.

This means:

- If one subspace mixes two different clusters → OT cost **strictly increases**.
- If a subspace specializes on a single cluster → OT cost **decreases**.

Thus, minimizing $\mathcal{L}_{\mathrm{OT}}$ encourages subspace specialization and discourages mixing of different semantics.

In summary, **OT naturally induces semantic specialization** because mixing distinct clusters is penalized more heavily than assigning each cluster to its own subspace. More details can be found in the appendix. Please refer to Appendix J for full details.

------------

**We have uploaded a revised version of the paper, which includes additional experiments, theoretical analysis, and various clarifications based on reviewer feedback. For ease of review, all revised text is highlighted in orange in the PDF.**


**Summary of Revisions**

- We provide a theoretical analysis explaining why OT induces subspace specialization in Theorem 3.1.
- We supplement a visualization of subspace assignment frequencies over FG/BG regions in Figure 5.
- We expanded Section 4.2 with a detailed analysis of the contributions of OT and the regularization term.
- We added an analysis of orthogonality versus specialization in Appendix F.
- We present a comparison with randomly initialized subspaces in Appendix G.

For all remaining questions, please refer to our individual responses.

---

### Author Response · Authors · 2025-12-02
**Rebuttal summary**

Dear ACs and Reviewers,

We sincerely thank you for your efforts in coordinating the review process and express our highest respect for your dedication, given the unexpected OpenReview incident. We provide here a concise **rebuttal summary**.

The main questions from reviewers are as follows:
- **Why OT enables textual subspaces to specialize** for different visual semantics?
- Whether the subspaces contain textual information?
- How do the subspaces differ from the initialized embeddings?
- **The roles of orthogonality regularization and OT** in textual subspace specialization.

To address these concerns, we summarize our clarifications and new analyses below:

- We provide a theoretical analysis of why minimizing the OT cost naturally leads to textual specialization in the general response above. In the revision, we include a formal explanation in **Theorem 3.1**, which shows that minimizing the OT objective penalizes subspace mixing and induces specialization. We further visualize the subspace assignment frequencies over **foreground (FG)** and **background (BG)** regions in Figure 5.

- Subsection **4.2 “What Makes the Textual Specialization”** has been expanded with additional analysis and visualizations.
  **Figure 5** illustrates subspace assignment frequencies across FG/BG regions and shows that **one subspace consistently attends to object foregrounds**, while the remaining subspaces focus on background regions, demonstrating emergent textual specialization.

- As shown in **Appendix G**, we compare **TokenCLIP** with **TokenCLIP-Visual**, where the latter uses only initialized embeddings without text information. In this case, TokenCLIP-Visual actually performs **visual–visual alignment** rather than cross-modal alignment, which limits its ability to generalize to unseen categories under zero-shot settings and highlights the necessity of text-aware subspaces.

- We provide a detailed analysis of how OT and orthogonality regularization contribute to specialization. The updated Figure 5 shows:
  1) **OT plays the dominant role** in enabling meaningful textual subspace specialization. Even without the regularization term, specialization emerges when OT is present.
  2) **When OT is removed**, the regularization term alone fails to induce meaningful specialization.
  3) The regularization term assists OT, further enhancing the specialization.

---

A revised version of the paper has been uploaded, with all updated content highlighted in **orange**.

Key revisions include:

- Added a **theoretical analysis** explaining why OT induces subspace specialization (**Theorem 3.1**).
- Added a visualization of FG/BG subspace assignments (**Figure 5**).
- Expanded **Section 4.2** with detailed analysis of OT and the regularization term.
- Added an analysis of **orthogonality versus specialization** (**Appendix F**).
- Added comparisons with **randomly initialized subspaces** (**Appendix G**).

Thank you once again for your time and consideration.

---

### Meta-Review · Area_Chair_GKgX · 2026-01-09

**Summary:**

This paper proposes a method to improve zero-shot anomaly detection of CLIP by using multiple orthogonal textual subspaces with optimal transport. Overall, some reviewers recognize the merits of the paper, e.g., fine-grained alignment among each patch features and multiple subspaces.  However, reviewers raised critical concerns regarding the fundamental justification of the "textual subspaces," as it is unclear if they retain textual semantics after optimization or simply become trained latent vectors. Major weaknesses include insufficient theoretical/experimental evidence for subspace semantics, missing implementation details (e.g., OT marginals), counterintuitive results (e.g., performance drop with greedy matching), and an incomplete explanation of the specialization mechanism, overly attributing it to OT rather than orthogonality regularization. Several minor errors like missing citations also weaken the manuscript.

**Reviewer Concerns:**

The reviewer concerns are partially addressed by the author's rebuttal and the feedback from reviewers. For example, the authors' rebuttal has addressed some specific clarifications regarding citation errors and typos, and performance discrepancy with the greedy matching variant (TokenCLIP-Van). However, some other concerns are not well addressed. For instance, the paper didnot well explain the fundamental working principle of the method, and how OT and orthogonality regularization contribute to the concept of semantic specialization.

**Reviewer Scores:**

After the rebuttal, the scores remained hovering around the average, accompanied by a reject opinion. Given the unresolved and inadequately explained issues, e.g., the intrinsic working principle, the reviewers generally are unlikely to strongly support acceptance of this paper. Based on these discussions and given the current high volume of submissions, the AC (Area Chair) leans towards rejection amid hesitation.

---

### Decision · Program_Chairs · 2026-01-26

Reject